# Compresso: Structured Pruning with Collaborative Prompting Learns Compact Large Language Models

## Abstract

Despite the remarkable success of Large Language Models (LLMs), the massive size poses significant deployment challenges, particularly on resource-constrained hardware. While existing LLM compression methods focus on quantization, pruning remains relatively unexplored due to the high cost of training-based approaches and data collection challenges. One-shot pruning methods, although cost-effective and data-free, have become dominant in LLM pruning, but lead to performance decline under the structured pruning setting. In this work, we introduce a new paradigm for structurally pruning LLMs, called *Compresso*. Our approach, through the collaboration of the proposed resource-efficient pruning algorithm and the LLM itself, learns optimal pruning decisions during the training process. Compresso addresses the challenges of expensive training costs and data collection by incorporating Low-Rank Adaptation (LoRA) into the $L_0$ regularization during the instruction tuning process. Then, we further augment the pruning algorithm by introducing a *collaborative prompt* that fosters collaboration between the LLM and the pruning algorithm, significantly boosting the overall performance. To this end, Compresso prunes LLaMA-7B to 5.4B, maintaining original performance and even surpassing LLaMA-7B in reading comprehension by 2.62%. Extensive experiments demonstrate that Compresso significantly outperforms one-shot pruning baselines across various sparsity ratios, achieving up to 2.21%, 11.43%, 7.04%, and 4.81% higher scores on the commonsense reasoning, reading comprehension, MMLU, and BBH benchmarks, respectively.

## 1 Introduction

The emergence of Large Language Models (LLMs) (Zhao et al., 2023; Chang et al., 2023; Brown et al., 2020) has revolutionized natural language processing tasks with remarkable success. However, their massive model size leads to the high inference costs. For example, GPT-3, with its 175B parameters (350GB in half-precision), requires a minimum of five A100 GPUs for inference. Consequently, LLM compression research has become pivotal in mitigating these high inference costs.

While existing LLM compression efforts focus on quantization (Liu et al., 2023; Xiao et al., 2023; Frantar et al., 2022; Yao et al., 2022), which reduces the bit number of model representations, the exploration of LLM pruning has remained limited. This is particularly true for *structured pruning*, which can directly cut inference costs on standard hardware but often is more challenging than unstructured pruning, as it strictly removes coherent groups of model parameters. A primary reason for the limited exploration on LLM pruning is that the success of various LLM families, such as GPT-3 (Brown et al., 2020), OPT (Zhang et al., 2022b), PALM (Chowdhery et al., 2022b), BLOOM (Scao et al., 2022), LLaMA (Touvron et al., 2023a), and LLaMA 2 (Touvron et al., 2023b) have demonstrated that increasing model size leads to enhanced capabilities. In contrast, the act of structured pruning, which reduces the model size, contradicts this trend and has been observed in existing attempt (Ma et al., 2023) to easily cause performance decline after pruning.

In this work, we explore the potential of structurally pruning non-essential parameters from LLMs as much as possible, while preserving their remarkable performance across various tasks. We begin by revisiting the existing pruning approaches. Unlike the best-performing approaches (Xia et al., 2022; Zhang et al., 2022a; Lagunas et al., 2021) used in the era of smaller models, which rely on a training-based process to compute full model parameter gradients, current efforts on LLM pruning all opt for one-shot pruning without any training (Frantar & Alistarh, 2023; Ma et al., 2023; Sun et al., 2023). This shift is driven by two primary factors. First, LLM training is exceptionally resource-intensive

due to its huge model size. Second, the training datasets for LLMs are extensive and often unavailable due to legal restrictions. Directly using open-sourced datasets can cause out-of-distribution issues, as the pruning data distribution is quite different with the pre-training. This leads us to fundamental questions *(i) If we can reduce the expensive training cost and find alternatives to training data, can training-based pruning offer a pathway to improve LLM pruning performance? (ii) Given the big disparities between small models and LLMs, are traditional pruning pipelines the optimal for LLMs?*

To this end, we introduce a new paradigm for structurally pruning Large Language Models called *Compresso*, which learns to make the optimal pruning decisions through a collaborative process involving a resource-efficient pruning algorithm and the target LLM itself. Compresso is built upon two key techniques. First, to address the challenges of high training costs and data collection in training-based pruning, we incorporate Low-Rank Adaptation (LoRA) (Hu et al., 2022) into $L_0$ regularization (Louizos et al., 2018) and use an instruction tuning dataset (Peng et al., 2023) as an alternative to training data. Specifically, we utilize learnable binary masks to decide whether to retain or prune each submodule (i.e., heads, FFN intermediate dimension, and hidden dimensions). Then, we employ $L_0$ regularization to optimize the mask values while concurrently updating model parameters through LoRA in the instruction tuning process. Furthermore, in contrast to one-shot LLM pruning methodologies, which often adopt a uniform sparsity ratio across all layers, Compresso automatically learns improved layer-wise sparsity ratios.

Second, different from existing approaches that treat the LLM as a passive role and subject them to various compression algorithms, our new pruning paradigm elevates LLMs to the role of a collaborative peer alongside pruning algorithms, leveraging the superiority and creativity of LLMs. To achieve this, we introduce a dedicated collaborative pruning prompt. This prompt explains the concept of pruning and its purpose, informs the LLM that it is undergoing pruning, and encourages the LLM to better adapt to the pruning process. We integrate this prompt into both the pruning and inference for the pruned LLM. Remarkably, this pruning prompt significantly boosts performance.

We summarize our key contributions as follows:

- We propose a novel paradigm for LLM pruning, called Compresso, where the LLM and a resource-efficient pruning algorithm collaboratively learn optimal pruning decisions during the instruction tuning. This paradigm showcases the vast potential of training-based LLM pruning and its superiority over one-shot pruning.

- We introduce two key techniques: a memory-efficient pruning algorithm incorporating LoRA and $L_0$ regularization, and a collaborative pruning prompt that encourages LLMs to better align with the pruning algorithm, significantly improving the pruning performance.

- Extensive experiments demonstrate that Compresso is able to prune LLaMA-7B to a 5.4B size, while maintaining its original generalization ability on zero-shot commonsense reasoning and reading comprehension, as well as few-shot MMLU and Big Bench Hard (BBH) benchmarks. Remarkably, Compresso-5.4B even surpasses LLaMA-7B in reading comprehension by 2.62%. Furthermore, across varying sparsity ratios, Compresso consistently outperforms one-shot pruning baselines on all benchmarks.

## 2 RELATED WORKS

**Compression of Small Language Models**. In the era of small language models (Devlin et al., 2018; Liu et al., 2019; Lan et al., 2019; Raffel et al., 2020), various compression techniques have been proposed to reduce the model size and inference costs, including weight pruning (Sanh et al., 2020b; Gordon et al., 2020; Zhang et al., 2022a; Xia et al., 2022), input token pruning (Li et al., 2023; Kim et al., 2022; Guan et al., 2022), quantization (Shen et al., 2020; Kim et al., 2021) and distillation (Sanh et al., 2020a; Jiao et al., 2020). We focus on weight pruning, particularly structured pruning, as it can directly reduce inference costs without special hardware support. Most state-of-the-art pruning methods involve a training process to update gradients and utilize them to estimate weight importance. Notable examples include CoFi (Xia et al., 2022) and nn pruning (Lagunas et al., 2021). However, these approaches cannot be directly applied to LLMs for two primary reasons. First, they are task-specific pruning methods requiring downstream training datasets. Therefore, the pruned models do not retain the generalization capabilities across different tasks. Second, the pruning process for LLMs demands substantial training resources (e.g., expensive GPU memory).

**Pruning Large Language Model.** Given the above challenges, training-based pruning for LLMs remains unexplored. Existing efforts, such as SparseGPT (Frantar & Alistarh, 2023), Wanda (Sun et al., 2023) and LLM-Pruner (Ma et al., 2023), all adopt low-resource, one-shot pruning methods without training. SparseGPT is the first unstructured pruning approach specifically developed to be fast enough for pruning LLMs within a few hours. Wanda applies magnitude pruning by weights and activations, which further improves the pruning speed than SparseGPT. Both can be extended for semi-structured pruning (i.e., the N:M sparsity (Pool & Yu, 2021; Hubara et al., 2021)). However, in practice, it is more challenging to translate the theoretically achieved sparsity in unstructured or semi-structured pruning to practical computation and storage savings on current GPU hardware (Frantar & Alistarh, 2023). LLM-Pruner (Ma et al., 2023) is the first attempt to structurally prune LLMs, offering the benefit of reducing both model computation and memory usage while keeping the overall LLM structure intact. It uses one-shot pruning based on first-order and approximated Hessian information and requires fine-tuning using LoRA to recover pruned model weights.

Despite its fast speed, one-shot pruning has limitations. First, it depends heavily on pre-defined weight importance metrics for pruning decisions, and thus adopts a uniform-sparsity ratio across all layers without considering the different redundancy at each layer. Second, error recovery for remaining model parameters is limited compared to training-based pruning, potentially affecting the final performance. Our Compresso addresses all these limitations.

Sheared LLaMA (Xia et al., 2023), a concurrent work, also adopts training-based pruning but demands a replicated pretraining dataset and 16 A100s with 80GB memory, which is cost-prohibitive for public users. In contrast, our method significantly reduces the substantial training resources, needing only $4 \times 32$GB V100s. Moreover, we introduce a novel collaborative pruning paradigm, which differentiates us from existing works.

**Prompting**. Prompting has emerged as a new paradigm for adapting pre-trained LLMs to new tasks by augmenting the model input with task-specific hints. Notable methods include template-based prompting (Schick & Schütze, 2021), instruction-based prompting (Wei et al., 2021; Sanh et al., 2022) , and Chain-of-Thought prompting (Wei et al., 2022). Despite its demonstrated success across a spectrum of NLP tasks (Chung et al., 2022; Goyal et al., 2022; Wei et al., 2022; Chowdhery et al., 2022a), the application of prompting for pruning LLMs remains unexplored in the literature.

**Instruction Tuning**. Fine-tuning LLMs with instructions enhances performance and generalization to unseen tasks (Wei et al., 2021; Ouyang et al., 2022; Chung et al., 2022). Self-Instruct (Wang et al., 2022) aligns LLMs to human intent by learning from instruction-following data generated by LLMs. Standford Alpaca (Taori et al., 2023) applies this strategy, producing 52k samples and fine-tuning the LLaMA model (Touvron et al., 2023a). Vicuna (Chiang et al., 2023) and GPT-4-LLM (Peng et al., 2023) further improve LLM performance by finetuning on either user-shared ChatGPT conversations or instruction-following data generated by GPT4. While LLM-Pruner uses instruction tuning to recover pruned LLMs' performance, pruning LLMs in instruction tuning has not been investigated.

## 3 METHODOLOGY

### 3.1 OVERVIEW

**Background and challenges**. Pruning LLMs using training-based methods is a nontrivial task with two key challenges. First, training-based pruning is resource-intensive, especially in terms of GPU memory. The process requires handling model parameters, their gradients, pruning masks, activations, and states in optimizers. For instance, pruning a LLaMA-13B model with the Adam optimizer requires at least 260GB of GPU memory, equivalent to 4 A100 GPUs. In practical training, due to the need for longer input lengths and larger batch sizes, the GPU memory requirement is much higher.

Second, it is crucial to preserve the generalization capability of LLMs after pruning. Therefore, dataset selection is crucial as a narrow or significantly different dataset from the original pre-training distribution may degrade performance. Despite training-based pruning's advantage over one-shot pruning in minimizing pruning errors, challenges arise due to the optimal weight updates on already converged LLMs and the complexity of replicating the original training setup. Consequently, learning the optimal pruning decisions remains a significant challenge.

**Overview**. Our approach, Compresso, addresses all the above challenges. Fig. 1 illustrates the overview of Compresso. First, Compresso utilizes the instruction tuning dataset as the pruning data in Sec. 3.2. Then, we incorporate LoRA and propose a memory-efficient pruning algorithm in Sec. 3.3. Finally, to achieve optimal pruning performance, we propose a new paradigm for pruning. Different

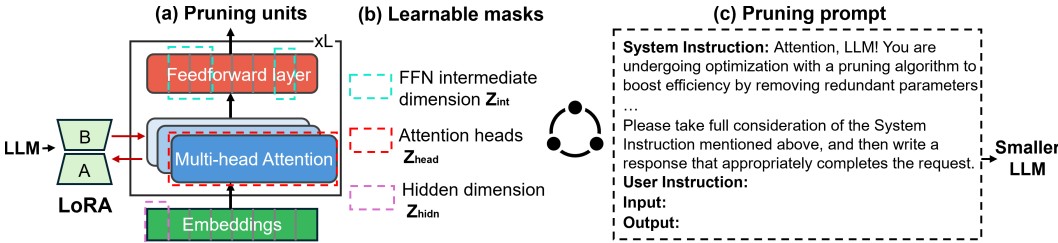

Figure 1: The overall framework of Compresso. We propose a collaborative pruning framework, where a memory-efficient pruning algorithm and target LLM work together through a collaborative prompt to learn optimal pruning decisions.

from conventional LLM compression pipeline (Frantar & Alistarh, 2023; Ma et al., 2023; Sun et al., 2023), Compresso leverages the superiority of LLM itself and designs a collaborative pruning process through a dedicated pruning prompt. We introduce this specially designed prompt in Sec. 3.4.

## 3.2 TRAINING DATA FOR PRUNING

Ideally, the distribution of pruning data should align with that of pre-training data, which typically comprises a large corpus of text from the Internet (Touvron et al., 2023a;b; OpenAI, 2023). LLMs learn to predict the next token in a sequence during pre-training. However, due to the limited access to the pre-training dataset, we explore the use of available public datasets as alternative resources.

Previous efforts typically sample a small subset of calibration data from the Crawled Corpus (C4) dataset (Raffel et al., 2019), which consists of clean English web text, and can be considered as a subset of pre-training data. However, while using C4 as pruning data yields reasonable perplexity, it performs poorly on zero-shot inference tasks (Liu et al., 2023). This is largely due to the different distributions between the C4 and the original pre-training data, leading to out-of-distribution issues.

We propose the use of instruction tuning datasets as pruning data. Despite their distribution differing from pre-training datasets, they have demonstrated success in fine-tuning pre-trained and converged LLMs to align with human intents. Specifically, we employ the GPT4-Alpaca dataset (Peng et al., 2023), which includes 52K GPT-4 generated instruction-following data in English.

## 3.3 EFFICIENT TRAINING-BASED STRUCTURED PRUNING

We now introduce our pruning algorithm designed to mitigate the substantial resources (i.e., the memory consumption) during training. The basic idea is: *(i)* we introduce a set of binary masks $\mathbf{Z} \in \{0, 1\}$ to indicate whether to drop ($\mathbf{Z} = 0$) or retain ($\mathbf{Z} = 1$) each masked submodule and thereby represent the remaining model size; *(ii)* we freeze the original LLM and utilize LoRA to inject extra trainable rank decomposition matrices into each layer of the LLM. This significantly reduces the number of trainable parameters and the required GPU memory; *(iii)* we jointly optimize these mask values and the LoRA modules using an augmented $L_0$ regularization (Louizos et al., 2018; Wang et al., 2020) method. This ensures the pruned model size meets the given constraints.

**Masking structured modules in LLMs**. We allow to prune three module types: attention heads, FFN intermediate dimensions, and hidden dimensions (i.e., the output dimensions of multi-head attention and FFN layers). Specifically, we mask attention heads by introducing variables $\mathbf{Z_{head}}^i \in \{0, 1\}$ to multi-head attention, where the $i^{th}$ head's corresponding $Q, K, V, O$ matrices are assigned the shared mask. We also allow for the pruning of fine-grained FFN intermediate dimensions by introducing $\mathbf{Z_{int}}^i \in \{0, 1\}^{d_f}$. To prune hidden dimensions, we follow CoFi (Xia et al., 2022) and define a set of masks $\mathbf{Z_{hidn}} \in \{0, 1\}^d$, shared across layers due to the residual connection between the same dimension in consecutive layers. Let $h \in \mathbb{R}^{n \times k}$ and $x \in \mathbb{R}^{n \times d}$ denote the original target module outputs and inputs. The training-based pruning can be formalized as the following:

$$h = \mathbf{Z_{head/int}} \cdot (W_0 x + \nabla W x) \cdot \mathbf{Z_{hidn}} \tag{1}$$

Where $W_0 \in \mathbb{R}^{d \times k}$ and $\nabla W \in \mathbb{R}^{d \times k}$ refer to a pre-trained weight matrix and its accumulated gradient updates. The above masks introduce a negligible number of extra trainable parameters. As shown in Table 1, LLaMA-7B and LLaMA-13B require only 0.35M and 0.56M masks respectively.

**Injecting LoRA modules**. In Equation 1, training-based pruning requires the updating of both model parameters and trainable masks. However, due to the massive size of LLMs, full gradient updates

on all parameters are very expensive. To address this, we incorporate lightweight LoRA (Hu et al., 2022) modules into the LLM, significantly reducing the training cost.

LoRA, due to its simplicity and effectiveness, has gained increasing attention in academia and industry and is widely used in fine-tuning LLMs in resource-limited scenarios (Hu et al., 2023; Gao et al., 2023). We apply LoRA in a novel manner. Formally, LoRA constrains gradient updates on all parameters via two low-rank matrices $A \in \mathbb{R}^{r \times k}$ and $B \in \mathbb{R}^{d \times r}$ ($r \ll min(d, k)$): $W_0 x + \nabla W x = W_0 x + BAx$. This allows for easy integration of LoRA with pruning. Equation 1 can be formalized as:

$$h = \mathbf{Z}_{\mathbf{head/int}} \cdot (W_0 x + \nabla W x) \cdot \mathbf{Z}_{\mathbf{hidn}} = \mathbf{Z}_{\mathbf{head/int}} \cdot (W_0 x + BAx) \cdot \mathbf{Z}_{\mathbf{hidn}} \quad (2)$$

Then, during our pruning process, we fix the original LLM parameters, with only the LoRA modules and pruning masks as trainable parameters. This allows Compresso to jointly update the gradient for pruning masks and model parameters through LoRA, thereby learning optimal pruning decisions. As shown in Table 1, the total trainable parameters are a minimal 4.54M and 7.11M for LLaMA-7B and LLaMA-13B, respectively.

|  | LLaMA-7B | LLaMA-13B |
|---|---|---|
| Masks | 0.35M | 0.56M |
| LoRA modules | 4.19M | 6.55M |
| Total | 4.54M | 7.11M |

Table 1: Required trainable parameters.

**Learning mask values with augmented L0 regularization.** Existing LLM pruning works (Frantar & Alistarh, 2023; Sun et al., 2023; Ma et al., 2023) rely on pre-defined weight importance metrics to decide on pruning or retaining weights, typically adopting a uniform-sparsity strategy. This approach, treating all layers equally and retaining the top $p$ important weights within each layer, can lead to suboptimal pruning due to varying redundancy levels across layers. LLM-Pruner manually identifies layers sensitive to pruning and excludes the first and final layers from pruning. In contrast, Compresso employs an automated approach, deciding the mask values via $L_0$ regularization without any weight importance scoring. During this process, it also learns to distribute sparsity across layers.

Let $\hat{s}$ represent the expected sparsity and $M$ denote the original full model size. We calculate the remaining model size based on the mask values $\mathbf{Z}_{\mathbf{head}}$, $\mathbf{Z}_{\mathbf{int}}$ and $\mathbf{Z}_{\mathbf{hidn}}$. Let $N_h$ be the number of attention heads, $L$ is the number of transformer layers, and $d_h$ and $d_f$ the dimension of each head and FFN intermediate size, respectively. The sparsity function is defined as follows:

$$\hat{s}(\mathbf{Z}) = \frac{1}{M} \cdot 4 \cdot d_h \cdot \sum_i^L \sum_j^{N_h} \sum_k^d \mathbf{Z}_{\mathbf{head}}^{(i,j)} \cdot \mathbf{Z}_{\mathbf{hidn}}^{(k)} + \frac{1}{M} \cdot 3 \cdot \sum_i^L \sum_j^{d_f} \sum_k^d \mathbf{Z}_{\mathbf{int}}^{(i,j)} \cdot \mathbf{Z}_{\mathbf{hidn}}^{(k)} \quad (3)$$

The two terms calculate the sparsity in attention heads and FFN layers, respectively. To learn the optimal mask values, we employ the $L_0$ reparameterization proposed by (Louizos et al., 2018), which enables differentiation of binary, non-differentiable masks $\mathbf{Z}$ using the hard concrete distribution:

$$\boldsymbol{u} \sim U(0, 1)$$
$$\mathrm{s} = \mathrm{sigmoid}((\log \frac{\boldsymbol{u}}{1 - \boldsymbol{u}} + \log \boldsymbol{\alpha})/\beta)$$
$$\tilde{s} = s \times (r - l) + l$$
$$\mathbf{Z} = \min(1, \max(0, \tilde{s}))$$
$$(4)$$

where $U(0, 1)$ is a uniform distribution in the interval [0,1]; $l < 0$ and $r > 0$ are two constants that stretch the sigmoid output into the interval $(l, r)$. $\beta$ is a hyperparameter that controls the steepness of the sigmoid function. We adopt the common practice of setting $l$ to -0.1, $r$ to 1.1 and $\beta$ to $\frac{2}{3}$ (Xia et al., 2022). $\boldsymbol{\alpha} = \{\alpha_j\}_{j=1}^{|\mathbf{Z}|}$ are the main learnable parameters. During training, the hard concrete parameters $\boldsymbol{\alpha}$ and $\boldsymbol{u}$ determine the values of masks $\mathbf{Z}$. We learn masks $\mathbf{Z}$ by updating these learnable parameters of the distributions from which the masks are sampled in the forward pass. Moreover, these learnable parameters and masks can be jointly optimized with the original model parameters through LoRA modules, resulting in better pruning performance.

To control the desired sparsity of pruned models, we follow (Xia et al., 2022; Wang et al., 2020) to replace the vanilla $l_0$ objective with a Lagrangian multiplier. Let S be the target sparsity, $\hat{s}(\mathbf{M})$ be the expected sparsity determined by the masks $\mathbf{Z}$ in Equation 3. We impose an equality constraint $\hat{s}(\mathbf{Z}) = $ S by introducing a penalty:

$$L_{0reg}(\mathbf{Z}) = \lambda_1 \cdot (\hat{s}(\mathbf{Z}) - \mathrm{S}) + \lambda_2 \cdot (\hat{s}(\mathbf{Z}) - \mathrm{S})^2 \quad (5)$$

where the masks $\mathbf{Z}$ are determined by hard concrete parameters $\boldsymbol{\alpha}$ and $\boldsymbol{u}$ in Equation 4. $\lambda_1$ and $\lambda_2$ are two hyperparameters, which will be automatically adjusted by AdamW optimizer. The full training objective is a combination of the next token prediction loss and the $L_{0reg}$ loss.

**System Instruction:** Attention, LLM! You are undergoing optimization with a pruning algorithm to boost efficiency by removing redundant parameters. Here's what you need to know:
**Improved Efficiency**: The pruning algorithm streamlines your operations, enhancing overall performance.
**Resource Utilization**: Reducing unnecessary parameters optimizes computational resources and memory requirements.
**Maintaining Quality**: The algorithm selectively prunes to preserve language generation capabilities while minimizing impact on performance.
**Iterative Refinement**: Pruning occurs gradually, with fine-tuning after each step to ensure a smooth transition.
**Collaboration**: Embrace changes, continue generating high-quality language outputs to contribute to the optimization process.
**Continuous Learning**: Pruning helps you evolve, focusing on essential aspects of language generation.
Together, we'll create a leaner and more efficient version of you. Let's unlock your full potential through pruning!
Please take full consideration of the System Instruction mentioned above, and then write a response that appropriately completes the request.
**User Instruction:**
**Input:**
**Output:**

Figure 2: An example to illustrate the use of our prompt in the proposed collaborative pruning.

### 3.4 PRUNING WITH COLLABORATIVE PROMPT

In this section, we introduce how our memory-efficient pruning algorithm and the LLM itself collaborate for pruning. Unlike traditional compression approaches where the target model plays a passive role, providing only performance metrics, our work introduces a paradigm shift by enabling LLMs to play an active, collaborative role through prompting. This fosters a collaborative environment where the target LLMs and pruning algorithms work together, significantly enhancing the pruning algorithms' ability to make optimal decisions.

This idea is inspired by the recent success achieved in various tasks by prompting LLMs (Sanh et al., 2022; Wei et al., 2022; Zhou et al., 2023). By adding a prompt (often optimized manually) before the inputs, LLMs can deliver competitive performance on many unseen tasks without the need of fine-tuning. The implication is that as long as LLM is appropriately instructed, it can perform well on downstream tasks it has never seen before. Consequently, a natural question arises: *Despite current LLMs not being trained on pruning tasks, can we design a pruning-dedicated prompt to instruct LLMs about the knowledge of pruning tasks and collaborate better with the pruning algorithm?*

Fig. 2 shows our dedicated pruning prompt and its utilization throughout the pruning process. Specifically, we adhere to three principles when designing the prompt: *(i)* inform the LLM that it is undergoing pruning by a pruning algorithm; *(ii)* explain the concept of pruning and its purpose; *(iii)* encourage collaboration between the LLM and the pruning algorithm. By following these principles, we utilize GPT4 to assist in crafting this prompt, which we refer to as the '*collaborative prompt*'.

During the pruning process, we place the prompt before the input text (as shown in Fig. 2). Following the practice of instruction tuning (Taori et al., 2023), we do not compute the next token generation loss for the prompt section. The collaborative prompt is used in both the pruning and inference stages.

## 4 EXPERIMENTS

### 4.1 SETTING

**Setup**. As introduced in Sec. 3.2, we use the GPT4-Alpaca dataset (Peng et al., 2023) as the pruning data, and empirically set a total of 7 epochs. The first epoch is a fine-tuning epoch, during which no pruning is performed. From the second to the fifth epoch, we follow a cubic sparsity schedule (Srinivas et al., 2022), gradually increasing the sparsity from 0 to the target ratio. In the final two epochs, we fix the sparsity and optimize the mask values under the fixed target sparsity. Once the pruning process is complete, we follow LLM-Pruner (Ma et al., 2023) to perform an additional two epochs of fine-tuning on the pruned model. We train using the AdamW optimizer, with a linear learning rate schedule, an initial learning rate of 5e-5, and a batch size of 8. The hyperparameters $\lambda_1$ and $\lambda_2$ from Equation 5 are automatically adjusted using the AdamW optimizer with a learning rate of 0.05. All experiments are conducted on 4 Nvidia V100 GPUs.

**Models and Evaluations**. We evaluate Compresso on the LLaMA (Touvron et al., 2023a) family. We prune LLaMA-7B to three different sparsity ratios, resulting in smaller models with 5.4B, 5B and 4.5B parameters. Unlike existing pruning works that only evaluate perplexity for next token prediction and commonsense reasoning tasks, we follow the original LLaMA families (Touvron et al., 2023a;b) to measure the effectiveness of pruned LLMs across three key application domains:

- **Zero-shot Commonsense Reasoning**. We evaluate the 0-shot results for 7 commonsense reasoning benchmarks: StoryCloze (Mostafazadeh et al., 2017), PIQA (Bisk et al., 2020),

Table 2: Zero-shot commonsense reasoning performance. Our pruned LLMs at 5.4B, 5B, and 4.5B retain 96%, 92%, and 90% of the original LLaMA-7B's capability, respectively.

| LLaMA-7B | Method | StoryCloze | PIQA | HellaSwag | WinoGrande | ARC-e | ARC-c | OBQA | Avg. |
|---|---|---|---|---|---|---|---|---|---|
| 7B | - | 78.35 | 78.67 | 56.98 | 70.01 | 75.29 | 41.81 | 34.2 | 62.19 |
| 5.4B | SparseGPT | 76.10 | 75.24 | 51.58 | 67.56 | 68.98 | 36.09 | 30.8 | 58.12 |
| | Wanda | 76.53 | 74.80 | 52.63 | 64.01 | **70.41** | 38.48 | 29.6 | 58.06 |
| | LLM-Pruner | 79.00 | **77.53** | 53.42 | 65.67 | 70.29 | 37.71 | 30.4 | 59.14 |
| | **Compresso** | **83.16** | 75.46 | **53.44** | **67.80** | 68.64 | **37.97** | **34.2** | **60.09** |
| 5.0B | SparseGPT | 73.97 | 73.23 | 46.99 | **65.58** | 66.03 | 33.27 | 27.2 | 55.18 |
| | Wanda | 73.49 | 71.38 | 47.84 | 61.16 | 65.06 | 33.44 | 28.8 | 54.45 |
| | LLM-Pruner | 77.28 | **75.63** | **50.78** | 65.19 | 63.55 | 33.36 | 28.8 | 56.37 |
| | **Compresso** | **79.10** | 73.07 | 49.16 | 64.80 | **66.20** | **37.20** | **29.8** | **57.05** |
| 4.5B | SparseGPT | 71.77 | 69.90 | 43.29 | **64.95** | 61.86 | 30.37 | 23.8 | 52.27 |
| | Wanda | 52.32 | 57.07 | 29.01 | 48.22 | 33.45 | 18.68 | 1.6 | 34.33 |
| | LLM-Pruner | 75.41 | **73.39** | 47.06 | 64.17 | 59.18 | 30.72 | 26.2 | 53.73 |
| | **Compresso** | **78.14** | 72.85 | **47.18** | 63.38 | **65.99** | **35.07** | **29.0** | **55.94** |

Table 3: Zero-shot performance comparison with one-shot pruning on reading comprehension.

| LLaMA-7B | Method | BoolQ | RACE-High | Avg. |
|---|---|---|---|---|
| 7B | - | 75.17 | 40.29 | 57.73 |
| 5.4B | SparseGPT | 71.13 | 37.50 | 54.32 |
| | Wanda | 73.48 | 37.60 | 55.54 |
| | LLM-Pruner | 63.21 | 34.64 | 48.92 |
| | **Compresso** | **79.08** | **41.63** | **60.35** |
| 5.0B | SparseGPT | 65.99 | 36.74 | 51.35 |
| | Wanda | 67.85 | 35.02 | 51.43 |
| | LLM-Pruner | 63.52 | 34.35 | 48.93 |
| | **Compresso** | **73.55** | **39.62** | **56.58** |
| 4.5B | SparseGPT | 64.52 | 36.26 | 50.39 |
| | Wanda | 54.34 | 24.59 | 39.47 |
| | LLM-Pruner | 62.69 | 32.73 | 47.70 |
| | **Compresso** | **68.69** | **36.36** | **52.52** |

HellaSwag (Zellers et al., 2019), WinoGrande (ai2, 2019), ARC easy and challenge (Clark et al., 2018), and OpenBookQA (OBQA) (Mihaylov et al., 2018).

- **Reading Comprehension**. We also evaluate the 0-shot performance on two reading comprehension benchmarks: BoolQ (Clark et al., 2019) and RACE-High (Lai et al., 2017).

- **Popular Aggregated Benchmarks**. Besides, we evaluate the in-context learning ability under a few-shot setting. We report the results on MMLU (5 shot) (Hendrycks et al., 2020), which consists of 57 tasks covering STEM, humanities, social science, etc, and Big Bench Hard (BBH) (3 shot) (Suzgun et al., 2022), which includes 23 challenging tasks.

For commonsense reasoning and reading comprehension, we use *lm-eval-harness* (Gao et al., 2021) to carry out the evaluations. For MMLU and BBH, we use *InstructEval* (Chia et al., 2023).

**Baselines**. Due to constraints such as the unavailability of open-source code or the need for substantial GPU resources, we setup two types of baselines: (i) SparseGPT and Wanda, initially designed for unstructured prunig and extendable to N:M sparsity. We further extend them for structured pruning; and (ii) LLM-Pruner, a representative one-shot structured pruning method. It's crucial to note that the commonsense reasoning metrics used in LLM-Pruner differ from other compression works, and different versions of the *lm-eval-harness* can cause numerical differences. For a fair comparison, we utilize the latest *lm-eval-harness* implementation for standard accuracy evaluation.

## 4.2 MAIN RESULTS

**Zero-shot Commonsense Reasoning.** Table 2 shows the zero-shot performance of pruned LLMs of varying sizes on commonsense reasoning. Compresso reduces the size of the original LLaMA-7B to 5.4B, 5B, and 4.5B, retaining 96%, 92%, and 90% of its commonsense reasoning capability, respectively. Interestingly, the pruned 5.4B and 5B models even surpass the original LLaMA-7B by 4.81% and 0.75% on StoryCloze, respectively. Compared to other one-shot pruning baselines, Compresso consistently scores higher across all sparsity ratios, particularly at higher sparsity ratios. For example, when pruned to 4.5B, Compresso significantly outperforms SparseGPT, Wanda, and LLM-Pruner by 3.67%, 21.72%, and 2.21%, respectively. Notably, Wanda faces a substantial accuracy

Table 4: Few-shot performance on MMLU and BBH.

| LLaMA-7B | Method | MMLU (5-shot) | | | | | BBH (3-shot) | | |
|---|---|---|---|---|---|---|---|---|---|
| | | Humans | STEM | Social | Other | Avg. | NLP | Algorithmic | Avg. |
| 7B | - | 34.3 | 32.3 | 40.6 | 40.9 | 36.80 | 36.60 | 28.93 | 32.34 |
| 5.4B | SparseGPT | 25.9 | 23.4 | 29.8 | 28.5 | 26.80 | 32.36 | 24.21 | 27.83 |
| | Wanda | 24.5 | 22.3 | 24.9 | 26.2 | 24.50 | 34.09 | 24.86 | 28.96 |
| | LLM-Pruner | 25.7 | 23.0 | 23.9 | 26.3 | 24.86 | 34.82 | 24.29 | 28.97 |
| | **Compresso** | **32.1** | **27.3** | **32.7** | **35.2** | **31.90** | **35.27** | **28.42** | **31.47** |
| 5.0B | SparseGPT | 25.8 | 23.8 | 25.6 | 25.3 | 25.21 | 32.71 | 26.75 | 29.40 |
| | Wanda | 19.0 | 26.4 | 25.6 | 24.5 | 23.32 | 27.35 | 22.86 | 24.86 |
| | LLM-Pruner | 21.7 | 23.9 | 22.2 | 23.2 | 23.22 | 29.45 | 24.08 | 26.46 |
| | **Compresso** | **28.3** | **26.4** | **27.0** | **28.6** | **27.68** | **35.95** | **27.53** | **31.27** |
| 4.5B | SparseGPT | 20.8 | 23.9 | 23.7 | 21.7 | 22.30 | 29.50 | 24.29 | 26.61 |
| | Wanda | 9.9 | 18.9 | 17.2 | 19.5 | 15.65 | 22.45 | 10.99 | 16.08 |
| | LLM-Pruner | 24.3 | 22.3 | 22.8 | 25.6 | 23.85 | 27.64 | 22.29 | 24.67 |
| | **Compresso** | **25.0** | **25.3** | **25.8** | **28.0** | **25.92** | **32.62** | **24.75** | **28.25** |

Table 5: Ablation study on using different training data in Compresso.

| | C4 Subset | LLM-QAT | GPT4-Alpaca |
|---|---|---|---|
| Commonsense Reasoning | 56.41 | 58.62 | **60.09** |
| Reading Comprehension | 52.78 | 55.18 | **60.35** |
| MMLU (5-shot) | 22.91 | 27.89 | **31.90** |
| BBH (3-shot) | 28.69 | 29.65 | **31.47** |

drop at higher sparsity ratios (4.5B). This could be due to Wanda's direct weight pruning approach, which relies on activation and weights.

**Zero-shot Reading Comprehension**. Table 3 compares the performance of pruned LLMs on reading comprehension. Remarkably, our pruned 5.4B model surpasses the original LLaMA-7B with 3.91% and 1.34% higher scores on BoolQ and RACE-High, respectively. This suggests a significant redundancy in LLaMA for reading comprehension. In contrast, all comparison baselines exhibit a decline in performance. Unlike in commonsense reasoning, LLM-Pruner performs poorly on this benchmark. For example, Compresso surpasses LLM-Pruner by **15.87%**, **10.03%**, and **6.0%** on BoolQ when pruned to 5.4B, 5B, and 4.5B, respectively. Similarly, on RACE-High, we surpass LLM-Pruner by **6.99%**, **5.27%**, and **3.63%** under the three target model sizes, respectively.

**Few-shot Evaluation on MMLU and BBH.** In context learning is a fundamental ability of LLMs (Brown et al., 2020). To verify whether the pruned LLMs retain the in context learning capability, we evaluate on the MMLU with 5-shot and BBH with 3-shot setting. As shown in Table 4, Compresso significantly outperforms LLM-Pruner on these few-shot benchmarks, with improvements of up to 7.04% and 4.81% on MMLU and BBH, respectively. Interestingly, LLaMA-7B shows more redundancy on BBH, allowing us to retain 96% of its capability while reducing the size from 7B to 5B. Despite the challenge of pruning on MMLU, when pruned to 5.4B, LLM-Pruner experiences a noticeable drop of **-11.94%** on MMLU, while Compresso retains 87% of LLaMA-7B's capability.

**Analysis.** In summary, we prove that Compresso can prune LLaMA-7B down to 5.4B, maintaining performance in both zero-shot and few-shot capabilities. In contrast, all other one-shot pruning baselines fail to maintain the original LLaMA-7B's generalization ability across all benchmarks. For instance, LLM-Pruner, while effective in commonsense reasoning, experiences significant performance drops in reading comprehension as well as few-shot MMLU and BBH. This occurs even though it also conducts instruction tuning to recover information loss for pruned models. This highlights the necessity of preserving key model weights during pruning, as fine-tuning cannot fully compensate for the loss of crucial information. Based on these observations, we encourage the LLM compression community to evaluate compressed LLMs on more essential benchmarks rather than solely reporting the zero-shot commonsense reasoning performance.

## 4.3 ABLATION STUDY

**The impact of pruning data**. In our experiments, we found that the dataset selection greatly impacts the final results of training-based pruning. We set two baselines, referencing prior works in LLM pruning and quantization: (1) C4 subset, a popular choice in many compression works, from which we sample more data for training. Specifically, we randomly sample 20k corpus of 1024 tokens from the C4 dataset. (2) LLM-QAT data, proposed by LLM-QAT (Liu et al., 2023), which begins with

Table 6: Ablation study on the removal of pruning prompt at different stages. **Blue** color indicates the performance degradation when compared to the use of the pruning prompt.

| Task | w/o in both stages | w/o in training | w/o in inference | | |
|------|--------------------|-----------------|------------------|--|--|
| | 5.4B | 5.4B | 5.4B | 5.0B | 4.5B |
| Commonsense Reasoning | 57.09 (-3.00) | 57.36 (-2.73) | 56.07 (-4.02) | 52.86 (-4.19) | 52.82 (-3.11) |
| Reading Comprehension | 55.84 (-4.51) | 56.62 (-3.73) | 56.52 (-3.83) | 51.39 (-5.19) | 48.51 (-4.01) |
| MMLU (5 shot) | 29.10 (-2.80) | 29.30 (-2.60) | 31.49 (-0.41) | 27.16 (-0.52) | 25.91 (-0.01) |
| BBH (3 shot) | 28.64 (-2.83) | 28.08 (-3.39) | 30.57 (-0.90) | 30.54 (-0.73) | 27.65 (-0.60) |

Table 7: Performance of pruned LLMs without post fine-tuning. **Blue** color indicates the performance drop while **Brown** indicates improvement compared to the performance after fine-tuning.

| Task | 5.4B | 5.0B | 4.5B |
|------|------|------|------|
| Commonsense Reasoning | 59.10 (-0.72) | 56.21 (-0.84) | 54.66 (-1.28) |
| Reading Comprehension | 58.11 (-2.24) | 56.07 (-0.51) | 51.51 (-1.01) |
| MMLU | 28.91 (-2.99) | 26.17 (-1.51) | 24.32 (-0.71) |
| BBH | 30.10 (-1.37) | 29.70 (-1.57) | 29.56 (+1.31) |

three randomly selected tokens from the vocabulary and uses the target LLM to generate the next token for a length of 1024. We follow the original setting to sample a total of 96k corpus.

Table 5 presents the results of Compresso pruning llama7b to 5.4B using three datasets. The results show that GPT4-Alpaca, an instruction tuning dataset, outperforms C4 and LLM-QAT's next token generation data, showcasing the importance of dataset choice in training-based pruning.

**The effectiveness of collaborative pruning**. In Compresso, the target LLM collaborates with the pruning algorithm for optimal pruning decisions. To evaluate the LLM role's effectiveness, we set up two experiments: *(i)* we exclude the pruning prompt from the training process, using it only during the final inference; *(ii)* we remove the pruning prompt only during the final inference stage; *and (iii) we remove the pruning prompt in both stages*. The results, as shown in Table 6, indicate that removing the pruning prompt at either stage significantly reduces the performance of pruned LLMs, particularly on commonsense reasoning and reading comprehension tasks. This demonstrates the effectiveness of our proposed collaborative pruning.

**The effectiveness of post fine-tuning.** Table 7 shows the benchmark of pruned LLMs without post fine-tuning. The results indicate that fine-tuning can slightly enhance performance, suggesting that Compresso preserves the most crucial model weights and effectively compensates for the information loss caused by pruning during training. This contrasts with LLM-Pruner, which heavily relies on post fine-tuning, with a big improvement on commonsense reasoning by up to 7.5%.

**Visualization and analysis**. Finally, we study the pruned LLM structures. When targeting the same 4.5B model, Fig. 3 (in Appendix) shows the remaining ratios of layer-wise heads and FNN intermediate sizes produced by our Compresso and LLM-Pruner. In contrast to LLM-Pruner, which adopts a uniform sparsity strategy for all middle layers while manually keeping the first and final layers unpruned, our Compresso automatically learns a different layer-wise sparsity ratio. Compresso tends to preserve more heads in the first and middle layers, it prune more heads in the final layers. For the FFN intermediate size, each layer is pruned by a similar number of parameters, but it can still be observed that the ratios of preserved FFN in the layers form a pattern resembling the letter "W". These findings suggest that the middle layers in LLM are also crucial for maintaining performance after pruning. Our superior results, as demonstrated in Table 2-4, suggest that the layer-wise sparsity ratio learned by Compresso is more effective in preserving the original LLM performance.

## 5 CONCLUSION

In this work, we propose Compresso, a collaborative structured pruning approach for large language models. Compresso addresses the challenges in training-based pruning by proposing a memory-efficient pruning algorithm that incorporates LoRA into $L_0$ regularization. Then, Compresso introduces a novel collaborative pruning paradigm where the pruning algorithm and target LLM work together through a collaborative prompt to learn the optimal pruning decisions during the instruction tuning process. Extensive experiments across diverse essential benchmarks demonstrate Compresso's superiority over existing one-shot LLM pruning works. Compresso can prune LLaMA-7B to a more compact 5.4B size while preserving its original zero-shot and few-shot generalization capabilities.

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

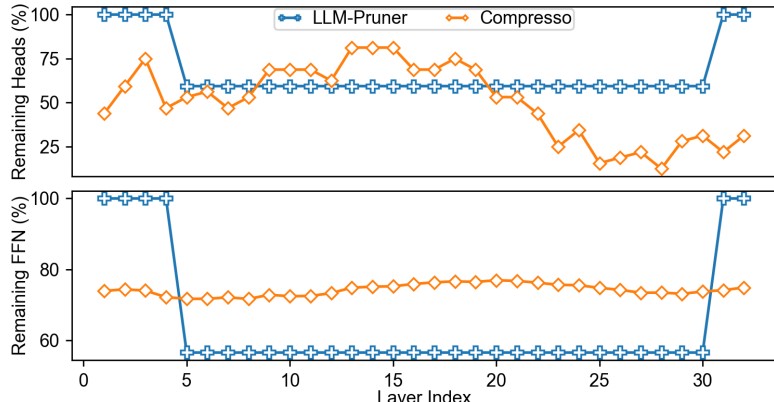

Figure 3: The remaining ratios of heads (upper) and FFN intermediate size (lower) among various layers when targeting a size of 4.5B.

Table 8: Zero-shot commonsense reasoning performance. By pruning LLaMA-13B down to 11B, Compresso preserves 100% of the original performance, and even surpasses the original LLama-13B.

| LLaMA-7B | Method | StoryCloze | PIQA | HellaSwag | WinoGrande | ARC-e | ARC-c | OBQA | Avg. |
|---|---|---|---|---|---|---|---|---|---|
| 13B | - | 79.58 | 77.8 | 57.05 | 72.69 | 76.64 | 45.82 | 32.8 | 63.2 |
| 11B | LLM-Pruner | 81.13 | **78.94** | **59.07** | 70.40 | 74.07 | 44.11 | 31.2 | 62.7 |
| | **Compresso** | **85.09** | 78.73 | 58.33 | **72.22** | **75.93** | **45.31** | **35.8** | **64.5** |

# A APPENDIX

## A.1 ADDITIONAL RESULTS

**Pruning LLaMA-13B**. We also evaluate Compresso on larger LLaMA-13B. As shown in Table 8, when pruning LLaMA-13B to 11B, Compresso not only preserves 100% of the original LLaMA-13B performance but also surpasses it on tasks such as StoryCloze, PIQA, HellaSwag, and OBQA. In contrast, the pruned model produced by LLM-Pruner shows a performance decline compared to the original 13B model.

**The generalization ability of our proposed collaborative pruning prompt on other sparse patterns**. To further validate the efficiency of our proposed pruning prompt, we broaden our experimental scope to include other sparse patterns such as unstructured and N:M sparse patterns. We utilize SparseGPT, a representative method for unstructured pruning and N:M sparsity, for the experiment. Specifically, we pruned LLaMA-7B under a 50% sparsity ratio for unstructured pruning and under N:M (2:4) sparsity using SparseGPT. We then conducted fine-tuning where our pruning prompt is integrated for the pruned LLMs.

We evaluate the pruned model under conditions with and without our pruning prompt for zero-shot commonsense reasoning and reading comprehension. As shown in Table 9, our pruning prompt significantly improves the performance on all benchmarks and tasks.

**Latency measurement**. We measure the inference latency of our pruned LLMs using the vllm framework (Kwon et al., 2023) on an A100. We randomly select 10 questions from the GPT4Alpaca dataset, and measure the inference latency of the generation process both with and without the use of pruning prompt. Table 10 shows the measured latency. All latency numbers represent end-to-end latency. Our pruned LLMs of different sizes all accelerate the token generation process of the original LLaMA-7B, both with and without the pruning prompt. Specifically, Compresso-5.4B and 4.5B accelerate the generation process by 1.17× and 1.19×, respectively. When we add our pruning prompt, the latency slightly increases, but compared to LLaMA-7B, there is still an acceleration of 1.10× and 1.13×.

**Detailed pruned architectures**. As shown in Table 11, we delve into the structures of our pruned LLaMA models across three different sizes. We draw three key observations: (i) across all three pruned models, Compresso exhibits a tendency to retain more heads in the first and middle layers, while pruning more heads in the final layers. This tendency, although less evident in the FFN intermediate, continues to persist. Notably, in the 5.4B model, the heads and FFN intermediate size

Table 9: Our collaborative pruning prompt can also improve the performance of pruned LLMs under the unstructured and N:M sparsity patterns.

| Method | Commonsense Reasoning | | | | | | | | Reading Comprehension | | |
|---|---|---|---|---|---|---|---|---|---|---|---|
| | StoryCloze | PIQA | HellaSwag | WinoGrande | ARC-e | ARC-c | OBQA | Avg. | BoolQ | RACE-High | Avg. |
| 2:4 sparsity | 75.13 | 70.18 | 45.52 | 58.33 | 55.85 | 31.4 | 23.6 | 51.56 | 67.00 | 36.17 | 51.59 |
| 2:4 sparsity (with prompt) | **76.16** | **71.00** | **47.56** | **60.85** | **59.30** | **33.7** | **29.2** | **53.97** | **69.11** | **37.99** | **53.55** |
| 50% unstructured | 79.10 | 71.93 | 48.87 | 62.04 | 60.44 | 34.13 | 25.6 | 54.59 | 67.19 | 37.13 | 52.16 |
| 50% unstructured (with prompt) | **79.69** | **73.12** | **50.58** | **64.33** | **66.2** | **36.86** | **30.0** | **57.25** | **70.95** | **37.22** | **54.09** |

Table 10: The serving latency on an Nvidia A100 using the vllm framework (Kwon et al., 2023).

| Model | Latency | Acceleration |
|---|---|---|
| LLaMA-7B | 195.58 ms | 1.0× |
| Compresso-5.4B | 170.76 ms | 1.17× |
| Compresso-5.4B (with Collaborative Pruning Prompt) | 180.81 ms | 1.10× |
| Compresso-4.5B | 167.93 ms | 1.19× |
| Compresso-4.5B (with Collaborative Pruning Prompt) | 177.16 ms | 1.13× |

of certain middle layers are completely preserved. (ii) Compresso generally prunes slightly more heads than FFN layers. (iii) Only a small number of hidden dimensions are pruned.

## A.2 LIMITATIONS AND DISCUSSIONS

We have demonstrated that our method, Compresso, can preserve $\geq 90\%$ of the original capabilities on LLaMA-7B and 13B. Our experiments also prove that we can perform structured pruning on LLaMA-33B using 4 V100s (each with 32GB memory). However, despite not encountering out-of-memory issues, due to limited resources and our small batch size (i.e., batch size=8), it requires a month of training time. Therefore, we have not evaluate the effectiveness of Compresso on larger model size, such as the 33B and 65B scales. However, our method is highly generalizable and can be scaled up to larger models in future research.

One limitation of our method is its heavy reliance on the quality of the dataset used for pruning. As discussed in Section 3.2, the original pre-training datasets for LLMs are typically non-public and require significant training resources. Currently, we use an instruction tuning dataset (i.e., GPT4alpaca) as the pruning data. Our experimental results suggest that an instruction dataset can be a suitable choice for pruning data, but it may not be the optimal choice. For example, we can prune LLaMA-7b to 4.5B while preserving ∼90% of its original performance. However, when we attempt to prune at a higher sparsity ratio, there is a noticeable performance decline. Addressing this issue typically requires the original pre-training dataset to compenstate for the information loss of the pruned LLMs. As such, the generation of a high-quality pruning dataset that closely aligns the distribution of original pre-training dataset is a crucial consideration. We believe it represents a pivotal area for future research in LLM pruning.

## A.3 CASE STUDY

We further evaluate the generalization ability of the pruned LLMs and the original LLaMA-7B by testing them on a human evaluation (Wang et al., 2022) dataset. As shown in Table 12, we provide some examples of the experimental results to highlight that the quality of the texts generated by our pruned LLaMA-5.4B is in no way inferior to that of the original LLaMA-7B. We found that the pruned model, produced by our Compresso, maintains the accuracy of generated texts and even produces more logical and reasonable expressions.

Table 11: The configurations of pruned LLaMA models.

| Model | Hidden | | #Head | #Intermediate |
|---|---|---|---|---|
| LLaMA-7B | 4096 | Layer1-32: | 32 | 11008 |
| Compresso-5.4B | 4042 | Layer 1-8:
Layer 9-16:
Layer 17-24:
Layer 25-32: | 19, 16, 32, 32, 21, 32, 32, 32
32, 32, 32, 30, 27, 32, 32, 32
27, 32, 32, 32, 26, 32, 6, 32
4, 10, 8, 6, 9, 8, 8, 8 | 7594, 7582, 11008, 11008, 7366, 11008, 11008, 11008
11008, 11008, 11008, 7298, 7514, 11008, 11008, 11008
7514, 11008, 11008, 11008, 7546, 11008, 7931, 11008
7943, 7899, 7843, 7731, 7899, 7896, 7978, 8116 |
| Compresso-5.0B | 4060 | Layer 1-8:
Layer 9-16:
Layer 17-24:
Layer 25-32: | 21, 22, 27, 25, 28, 25, 29, 24
26, 29, 30, 30, 30, 26, 31, 31
29, 28, 29, 25, 26, 22, 16, 19
19, 16, 16, 11,16, 14, 12, 13 | 8072, 8161, 8141, 7993, 7965, 7989, 7959, 7903
8001, 7933, 7862, 8055, 8133, 8278, 8260, 8391
8429, 8447, 8412, 8492, 8400, 8408, 8360, 8365
8277, 8235, 8155, 8120, 8153, 8173, 8184, 8255 |
| Compresso-4.5B | 4044 | Layer 1-8:
Layer 9-16:
Layer 17-24:
Layer 25-32: | 14, 19, 24, 15, 17, 18, 15, 17
22, 22, 22, 20, 26, 26, 26, 22
22, 24, 22, 17, 17, 14, 8, 11
5, 6, 7, 4, 9, 10, 7, 10 | 8136, 8184, 8145, 7942, 7889, 7890, 7935, 7886
8014, 7970, 7979, 8067, 8237, 8266, 8279, 8351
8401, 8433, 8415, 8466, 8446, 8391, 8324, 8312
8228, 8169, 8078, 8083, 8047, 8115, 8159, 8240 |

| **Original Model (LLaMA-7B)** | **Pruned Model (5.4B)** |
|---|---|
| Suggest a recipe for a vegan dessert. 1 cup of sugar, 1 cup of flour, 1 cup of water, 1 cup of milk, 1 cup of butter, 1 cup of eggs, 1 cup of vanilla, 1 cup of chocolate chips, 1 cup of nuts, 1 cup of raisins, 1 cup of coconut, 1 cup of cinnamon, 1 cup of baking powder, 1 cup of baking soda, 1 cup of salt, 1 cup of cocoa, 1 cup of vanilla, 1 cup of chocolate chips, 1 cup of nuts, 1 cup of raisins, 1 cup of coconut, 1 cup of cinnamon... | Suggest a recipe for a vegan dessert. One vegan dessert recipe that is easy to make and delicious is chocolate ice cream. Simply combine 1 cup of unsweetened organic milk, 1 cup of organic heavy cream, 1 teaspoon of pure vanilla extract, and 1 tablespoon of pure sugar in a saucepan. Heat over medium heat until the mixture is warm and the sugar has dissolved. Remove from heat and stir in 1 cup of unsalted nuts... |
| You need to write an email to negotiate your salary. I am writing to you to discuss my salary. I am currently working as a ______ in your company. I have been working here for ______ years. I am very happy with my job and I am very satisfied with the salary I am getting. I am very interested in continuing my job in your company. I would like to discuss my salary with you. I would like to ask for a ______ in my salary. I am very interested in... | You need to write an email to negotiate your salary. Subject: Negotiating Your Salary\n\nDear [Name of Hiring Manager],\n\nI am writing to you today to discuss the possibility of negotiating my salary. I have been working for [Company Name] for [Number of Years] and have been very happy with my job and my work. However, I have noticed that my salary is not at the same level as my colleagues who have been here... |
| You can easily make your holidays more eco-friendly by following these simple steps: 1. 2. 3. 4. 5. 6. 7. 8. 9. 10. 11. 12. 13. 14. 15. 16. 17. 18. 19. 20. 21. 22. 23. 24. 25. 26. 27. 28. 29. 30. 31. 32. 33. 34. 35. 36. 37. 38. 39. 40. 41. 42. 43. 44. 45. 46. 47. 48. 49. 50. 51. 52. 53. 54. 55. 56. 57. 58. 59. 60. 61. 62. 63. 64. 65. 66... | You can easily make your holidays more eco-friendly by following these simple steps: Yes, it is possible to make your holidays more eco-friendly by following these simple steps:\n1. Choose a destination that is close to home and avoid long flights.\n2. Travel by train or bus instead of by car.\n3. Use public transportation or ride a bicycle to get around.\n... |
| Give students tips on how to keep their nerves under control during class presentations. 1. Have students practice their presentations in front of a mirror. 2. Have students practice their presentations in front of a friend. 3. Have students practice their presentations in front of a group of friends. 4. Have students practice their presentations in front of a group of strangers. 5... | Give students tips on how to keep their nerves under control during class presentations. One way to keep your nerves under control is to practice your presentation a lot. This will help you feel more confident and comfortable with your material. Another way is to take a deep breath and focus on your audience. Remember that they are there to support you and help you succeed... |

Table 12: Some examples generated by original model and our pruned model.

