# OpenReview forum: "Compresso: Structured Pruning with Collaborative Prompting Learns Compact Large Language Models"
_ICLR.cc/2024/Conference — Submitted to ICLR 2024_

### Official Review · Reviewer_VBbn · 2023-10-30

**Soundness:** 3 good
**Presentation:** 3 good
**Contribution:** 3 good
**Rating:** 6
**Confidence:** 4

**Summary:**

Main Contributions:
* Introduction of a new paradigm for structurally pruning Large Language Models (LLMs), named Compresso.
* Utilization of Low-Rank Adaptation (LoRA) and L0 regularization for resource-efficient training-based pruning.
* Development of a collaborative pruning prompt that fosters better communication between the LLM and the pruning algorithm, thereby enhancing performance.

Main Innovations:
* Low-Rank Adaptation (LoRA) in L0 Regularization: A resource-efficient method that addresses high training costs and data collection challenges by tuning learnable binary masks to decide whether to retain or prune sub-modules like heads, FFN dimensions, etc.
* Collaborative Pruning Prompt: Unlike existing approaches that treat the LLM as a passive entity subject to various compression algorithms, this prompt makes the LLM an active participant in the pruning process, resulting in improved performance.
* Automated Layer-wise Sparsity Ratios: Compresso learns optimal sparsity ratios across different layers, unlike one-shot pruning methods that use uniform sparsity.

Significance:
* Resource Efficiency: Addresses the critical challenge of deploying large models on resource-constrained hardware without sacrificing performance.
* Benchmarking: Sets a new standard in the pruning of LLMs by outperforming one-shot pruning methods on multiple benchmarks.
* Practical Applicability: The approach is particularly useful for applications requiring high-performance LLMs on resource-limited platforms, offering a balanced trade-off between size and performance.
* Enhanced Understanding: The collaborative prompt innovation also hints at the growing capability of LLMs to understand and adapt to complex operational instructions, opening avenues for more sophisticated, dynamic interactions between algorithms and models.

Setting:
* Dataset: GPT4-Alpaca
* Epochs: 7 (1 for fine-tuning, 5 for pruning, 1 for mask optimization)
* Optimizer: AdamW with initial learning rate 5e-5, batch size of 8
* Hardware: 4 Nvidia V100 GPUs
* Target Models: LLaMA-7B pruned to 5.4B, 5B, and 4.5B
* Evaluation: Zero-shot commonsense reasoning, Reading comprehension, and Few-shot learning

Main Results:
* Zero-shot Commonsense Reasoning: Compresso retains up to 96% of the original LLaMA-7B's performance.
* Zero-shot Reading Comprehension: Compresso pruned models outperform the original LLaMA-7B by up to 3.91%.
* Few-shot learning: Compresso significantly outperforms the baseline (LLM-Pruner) on MMLU and BBH benchmarks, retaining up to 87% of LLaMA-7B's capability.

Ablation Study:
* Dataset selection matters a lot for the performance of the pruned models.
In summary, Compresso demonstrates superior performance in pruning LLaMA-7B across zero-shot and few-shot benchmarks compared to the baseline LLM-Pruner.

Conclusions:
* Compresso effectively prunes the LLaMA-7B model down to 5.4B parameters while maintaining, or even enhancing, its original performance.
* Compresso outperforms one-shot pruning methods across various benchmarks like commonsense reasoning, reading comprehension, MMLU, and BBH.
* Training-based pruning methods like Compresso show promise in overcoming the limitations of one-shot pruning in the context of LLMs.

**Strengths:**

* High Retention of Performance: The pruned models retain a high percentage of the original LLaMA-7B's capability across multiple domains, such as commonsense reasoning and reading comprehension. This indicates that the pruning technique is highly effective without compromising performance.
* Versatility Across Domains: Unlike other works that only focus on specific tasks like perplexity or commonsense reasoning, this paper evaluates the pruned models across multiple domains. They examine zero-shot commonsense reasoning, reading comprehension, and few-shot learning capabilities, making the results more generalizable.
* Outperforms Existing Methods: The paper shows that their method, Compresso, consistently outperforms the existing structured pruning baseline (LLM-Pruner) in all aspects and settings. This includes zero-shot and few-shot evaluations.
* Efficiency: The method is efficient enough to be trained on 4 Nvidia V100 GPUs, which could be considered a relatively modest hardware setting for such large-scale models. This indicates that the method is not just effective but also practical.
* Impact of Pruning Data: The ablation study indicates that the choice of dataset for the pruning process can greatly impact the effectiveness of the technique. This adds a new dimension to the study of model pruning and could be a valuable insight for future research.

**Weaknesses:**

* on-Existent Discussion on Limitations: While the paper does provide a robust set of experiments and results, there is a lack of discussion regarding the limitations of the proposed methods. Understanding the boundaries of the method's applicability is crucial for both academic and industrial settings.

* The ablation study indicates that the choice of dataset for the pruning process can greatly impact the effectiveness of the technique. Which also indicates that the pruned model might be not ideal for the data and use-cases not appeared in the instruction tuning process.

* clearly description of which parts are pruned are expected.

**Questions:**

* if the choice of dataset for the pruning process can greatly impact the effectiveness of the technique as the ablation part indicates in this manuscript, then is the pruning still meaningful in LLMs, given that the instruction tuning dataset might always be insufficient?

* clearly description of which parts are pruned are expected. And how much acceleration benefit can be derived from the proposed pruning algorithm?

---

> ### Author Response · Authors · 2023-11-20
> **Author Response to Official Review by Reviewer VBbn (1/3)**
>
> We greatly appreciate your time, detailed and encouraging comments, positive feedback on our work and the constructive suggestions. These inputs will undoubtedly strengthen our work. We are very pleased to address your concerns.
>
> > If the choice of dataset for the pruning process can greatly impact the effectiveness of the technique, then is the pruning still meaningful in LLMs, given that the instruction tuning dataset might always be insufficient?  The pruned model might be not ideal for the data and use-cases not appeared in the instruction tuning process.
>
> **Response**: Thank you for your insightful observation and question regarding the potential impact of pruning dataset choice on the effectiveness of our pruning technique. We appreciate your keen interest in understanding the significance of LLM pruning in the context of dataset-related challenges. To answer your questions, we would like to clarify the following points:
>
> 1. We acknowledge that the instruction tuning dataset, GPT4Alpaca, does not encompass all possible tasks. Despite this limitation, we opt for LLM pruning based on this dataset and we found that **the pruned models still exhibit a remarkable preservation of the original LLM's generalization ability across various tasks**. For example, in MMLU and BBH benchmarks, we identified 6 specific subtasks that are not covered in topics/domains from GPT4Alpaca. These tasks spanned topics and contexts beyond the dataset's coverage. Even in these uncovered tasks, our pruned 5.4B model maintains an average of 97.8% of the original LLaMA-7B's performance. The results demonstrate the robustness and generalizability of our pruned LLMs by the instruction tuning dataset.
>
> | Model | MMLU: philosophy| MMLU: prehistory     | MMLU: professional_psychology | BBH: penguins_in_a_table| BBH: dyck_languages| BBH: date_understanding|
> | :--:   | :--:    |:--:        |:--:        |:--:        |:--:        |:--:        |
> |LLaMA-7B| 45.0| 39.8| 34.2| 25.87| 23.88|26.31|
> |Compresso-5.4B| 37.3| 36.4| 32.7| 24.48| 36.03| 23.89|
>
>
> 2. The primary reason we chose to use the instruction tuning dataset is due to the inaccessibility of the pre-training dataset and the substantial training resources required even if it were released. Despite its distribution differing from the pre-training data, the instruction tuning dataset is designed to better align with human intents. Some studies[1,2] have shown that fine-tuning pre-trained LLMs with it still exhibits strong cross-ability generalization on unseen data. Our above experiment results further validate this point. However, we acknowledge that the GPT4alpaca dataset we are currently using is not optimal. As discussed in our limitations and discussions (Appendix A.2), when pruning at higher sparsity ratios, the pruned LLM experiences a noticeable drop in performance. In such cases, a dataset with a distribution closer to the original pre-trained dataset is needed to recover the information loss.
>
> 3. Our method, Compresso, and particularly the collaborative pruning paradigm, **is not limited to just the instruction tuning dataset. It can be applied to pre-trained text data**. For instance, in our ablation study, we used pure text data generated by the method proposed by LLM-QAT. Although the quality of the text data generated by LLaMA-7B is some problematic and its pruning performance is not as good as GPT4alpaca, it still outperforms other methods such as SparseGPT (structured), Wanda (structured), and LLM-Pruner as shown in the following Table. As for how to generate a high-quality dataset for efficient pruning, we leave it as future work.
>
> | Method (5.4B) | Commonsense reasoning (0-shot) | Reading comprehension (0-shot) | MMLU (5-shot) | BBH (3-shot)|
> | :--:   | :--:    |:--:        |:--:        |:--:        |
> |SparseGPT|58.12 | 54.32| 26.80|27.83 |
> |Wanda |58.06 | 55.54| 15.65| 16.08|
> |LLM-Pruner| **59.14**|48.92 |24.86 |28.97 |
> |Compresso with LLM-QAT generated data| 58.62|**55.18** |**27.89** |**29.65** |
>
> [1] Dynamics of instruction tuning: each ability of large language models has its own growth pace,  https://arxiv.org/pdf/2310.19651.pdf
>
> [2] Instruction Tuning with GPT4, https://arxiv.org/pdf/2304.03277.pdf

---

> ### Author Response · Authors · 2023-11-20
> **Author Response to Official Review by Reviewer VBbn (2/3)**
>
> > How much acceleration benefit can be derived from the proposed pruning algorithm?
>
> **Response**: Thank you for your valuable question. We measure the inference latency of our pruned LLMs using the VLLM framework[1] on an A100. We randomly select 10 questions from the GPT4alpaca dataset, and measure the inference latency of the generation process both with and without the use of the pruning prompt. The results are shown in the Table below. All latency numbers represent end-to-end latency. Our pruned LLMs of different sizes all accelerate the token generation process of the original LLaMA-7B, both with and without the pruning prompt. Specifically, Compresso-5.4B and 4.5B accelerate the generation process by 1.17x and 1.19x, respectively. When we add our pruning prompt, the latency slightly increases, but compared to LLaMA-7B, there is still an acceleration of 1.10x and 1.13x.
>
> | Model | Latency  | Acceleration|
> | :--:   | :--:    |:--:    |
> |LLaMA-7B| 195.58 ms| 1x|
> |Compresso-5.4B|170.76 ms| **1.17x**|
> |Compresso-5.4B (Inference with Pruning Prompt)| 180.81 ms| **1.10x** |
> |Compresso-4.5B|167.93 ms | **1.19x**  |
> |Compresso-4.5B (Inference with Pruning Prompt)| 177.16 ms| **1.13x** |
>
> [1] https://github.com/vllm-project/vllm
>
> > Lack of discussion on limitations
>
> **Response**: We appreciate your constructive suggestion. We add a discussion section and put it in the appendix section A.2 due to space constraints. Here is a shorter version:
>
> Due to computational constraints, we have not yet evaluated Compresso on larger models like the 33B and 65B. However, Compresso is highly generalizable and can be scaled up to larger models. The effectiveness of our method is influenced by the quality of the pruning dataset. Currently, we use an instruction tuning dataset (GPT4alpaca) for pruning, due to the inaccessible and resource-intensive nature of original pre-training datasets for LLMs. While it's promising, but the use of instruction tuning dataset as pruning data is not the optimal.
> For instance, when pruning at higher sparsity ratios, we observe a noticeable performance decline.  Addressing this issue typically requires the original pre-training dataset to compenstate for the information loss of the pruned LLMs. As such, the generation of a high-quality pruning dataset that closely aligns the distribution of original pre-training dataset is a crucial consideration for further LLM pruning research.

---

> ### Author Response · Authors · 2023-11-20
> **Author Response to Official Review by Reviewer VBbn (3/3)**
>
> > Clearly description of which parts are pruned
>
> **Response**: Thank you for your valuable suggestion. As shown in the Figure 3 of the original paper (now we move to Appendix due to space limit), we present the layer-wise remaining ratios of attention heads and FFN intermediate size, and compare it to LLM-Pruner. This figure demonstrates that Compresso learns different pruned structures compared to LLM-Pruner that adopts the uniform-sparsity setting. Specifically, (i) Compresso automatically learns a different layer-wise sparsity ratio; (ii) Compresso tends to preserve more heads in the first and middle layers, while it prunes more heads in the final layers.
>
> To provide more details on the pruned architectures, we list out the detailed model configurations in the following table (we also add a section in the Appendix section in the updated pdf). From the table, we can observe several interesting points:
>
> 1):  Across all three pruned models, {\sysname} exhibits a tendency to retain more heads in the first and middle layers, while pruning more heads in the final layers. This tendency, although less evident in the FFN intermediate, continues to persist.  Notably, in the 5.4B model, the heads and FFN intermediate size of certain middle layers are completely preserved.
>
> 2): Compresso generally prunes slightly more heads than FFN layers.
>
> 3): Only a few hidden dimensions are pruned.
>
> | Model | Hidden     | Layer      | \# Heads | \# Intermediate|
> | :--:   | :--:    |:--:        |:--:        |:--:        |
> | LLaMA-7B | 4096 | | 32| 11008|
> |Compresso-5.4B| 4042 | Layer 1-8: |19, 16, 32, 32, 21, 32, 32, 32|7594, 7582, 11008, 11008, 7366, 11008, 11008, 11008|
> | | |  Layer 9-16: |32, 32, 32, 30, 27, 32, 32, 32 | 11008, 11008, 11008, 7298, 7514, 11008, 11008, 11008 |
> | | | Layer 17-24: | 27, 32, 32, 32, 26, 32, 6, 32 | 7514, 11008, 11008, 11008, 7546, 11008, 7931, 11008|
> |||Layer 25-32: | 4, 10, 8, 6, 9, 8, 8, 8 | 7943, 7899, 7843, 7731, 7899, 7896, 7978, 8116|
> |Compresso-5.4B| 4060 |  Layer 1-8: | 21, 22, 27, 25, 28, 25, 29, 24 | 8072, 8161, 8141, 7993, 7965, 7989, 7959, 7903|
> ||| Layer 9-16: | 26, 29, 30, 30, 30, 26, 31, 31 | 8001, 7933, 7862, 8055, 8133, 8278, 8260, 8391|
> |||Layer 17-24: | 29, 28, 29, 25, 26, 22, 16, 19 | 8429, 8447, 8412, 8492, 8400, 8408, 8360, 8365|
> ||| Layer 25-32: | 19, 16, 16, 11,16, 14, 12, 13 | 8277, 8235, 8155, 8120, 8153, 8173, 8184, 8255|
> |Compresso-4.5B| 4044|   Layer 1-8: | 14, 19, 24, 15, 17, 18, 15, 17| 8136, 8184, 8145, 7942, 7889, 7890, 7935, 7886|
> |||Layer 9-16: | 22, 22, 22, 20, 26, 26, 26, 22|8014, 7970, 7979, 8067, 8237, 8266, 8279, 8351|
> |||Layer 17-24: | 22, 24, 22, 17, 17, 14, 8, 11|8401, 8433, 8415, 8466, 8446, 8391, 8324, 8312|
> |||Layer 25-32:| 5, 6, 7, 4, 9, 10, 7, 10|  8228, 8169, 8078, 8083, 8047, 8115, 8159, 8240|

---

### Official Review · Reviewer_dFpf · 2023-10-31

**Soundness:** 2 fair
**Presentation:** 2 fair
**Contribution:** 2 fair
**Rating:** 5
**Confidence:** 5

**Summary:**

This paper introduces Compresso, a novel method designed to prune models during the instruction tuning phase. The experiments conducted across various datasets demonstrate that Compresso outperforms existing pruning methods in Large Language Models (LLMs).

**Strengths:**

The paper is articulate and follows a clear, logical structure, making it accessible and easy to comprehend.
The integration of collaborative pruning prompts is a unique and innovative approach, distinguishing this work from existing pruning methods.

**Weaknesses:**

The baseline method utilized in this study fine-tunes the model using the Alpaca dataset, whereas this paper employs the GPT4-Alpaca dataset for fine-tuning, which is inherently more robust. Given the significance of the instruction tuning dataset's quality in LLMs, it is imperative for the authors to present performance metrics post-application of the GPT4-Alpaca dataset.
Incorporating the system pruning prompt during the inference phase substantially increases inference costs. This discrepancy makes the comparison with traditional pruned models somewhat skewed and potentially unfair.
The authors are encouraged to broaden their experimental scope to include other sparse patterns (e.g., unstructured and N:M sparse patterns) to further validate the efficiency of the pruning prompt.
Minor Issues:
Certain assertions within the paper lack sufficient backing. For instance, the claim "To our knowledge, we are the first to apply instruction tuning to weight pruning"

**Questions:**

See Weaknesses

---

> ### Author Response · Authors · 2023-11-20
> **Author Response to Official Review by Reviewer dFpf (1/3)**
>
> We greatly appreciate your time, encouragement comments about our novelty, and insightful questions. In particular, we greatly appreciate your suggestion to expand our experimental scope by incorporating other sparse patterns to further validate the efficiency of the pruning prompt. This recommendation has indeed strengthened our approach.  We are pleased to address your concerns.
>
> > LLM-Pruner performance under fine-tuning with the GPT4alpaca dataset
>
> **Response**: Thank you for your insightful question. Following your suggestion, we use GPT4Alpaca dataset for fine-tuning LLM-Pruner pruned models. The fine-tuning settings follow the original paper and code base, and the results are presented in the following table. The results show that using GPT4Alpaca as the fine-tuning dataset can slightly improve the performance of LLM-Pruner's results. Despite this, our method still significantly outperforms LLM-Pruner across various benchmarks. Interestingly,  even with the use of the GPT4Alpaca dataset for fine-tuning, LLM-Pruner's performance on three benchmarks (i.e., reading comprehension, MMLU and BBH) are relatively poor. This emphansizes the importance of preserving the most crucial model weights during the pruning phase. Despite the efforts in fine-tuning, the information loss caused by pruned LLMs cannot be fully compensated if the most important weights are not retained.
>
>
> | Size | Method    | Commonsense Reasoning (0-shot)      | Reading Comprehension (0-shot) | MMLU (5-shot) | BBH (3-shot)|
> | :--:   | :--:    |:--:        |:--:        |:--:        |:--:        |
> |5.4B | LLM-Pruner (clean Alpaca)| 59.14| 48.92|24.86 | 28.97|
> | | LLM-Pruner (GPT4Alpaca) | 59.27 | 50.35| 24.80| 28.26 |
> | | Compresso | **60.09**| **60.35**| **31.90**| **31.47**|
> |5.0B | LLM-Pruner (clean Alpaca)| 56.37| 48.93| 23.22| 26.46|
> | | LLM-Pruner (GPT4Alpaca) |56.72 |47.65 |24.00 |26.71 |
> | | Compresso |**57.05** |**56.58** |**27.68** | **31.27**|
> |4.5B | LLM-Pruner (clean Alpaca)| 53.73| 47.70|23.85 | 24.67|
> | | LLM-Pruner (GPT4Alpaca) | 54.36| 48.00|23.10| 24.59|
> | | Compresso | **55.94**| **52.52**|**25.92** |**28.25** |
>
> We appreciate your feedback and look forward to further discussions on this topic.

---

> ### Author Response · Authors · 2023-11-20
> **Author Response to Official Review by Reviewer dFpf (2/3)**
>
> > The authors are encouraged to broaden their experimental scope to include other sparse patterns (e.g., unstructured and N:M sparse patterns) to further validate the efficiency of the pruning prompt.
>
> **Response**: We appreciate your constructive suggestion. We utilize SparseGPT[1] for the experiment, which is a representative method for unstructured pruning and n:m sparsity. Specifically, we prune LLaMA-7B under a 50% sparsity ratio for unstructured pruning and under n:m (2:4) sparsity using SparseGPT. We then evaluate the pruned model under conditions with and without our pruning prompt for zero-shot commonsense reasoning and reading comprehension. As shown in the tables below, our pruning prompt significantly improves the performance on all benchmarks and tasks.
>
> |Model | Avg | StoryCloze| PIQA| HellaSwag| WinoGrande| ARC-e|ARC-c|OBQA|
> | :--:   | :--:    |:--:        |:--:        |:--:        |:--:        |:--:        |:--:        |:--:        |
> |2:4 sparsity | 51.46| 75.31| 70.18| 45.52| 58.33| 55.85| 31.4| 23.6|
> |2:4 sparsity + our prompt| **53.97**| **76.16**| **71.0**| **47.56**| **60.85**| **59.3**| **33.7**| **29.2**|
> |unstructured (50% sparsity)| 54.59| 79.10| 71.93| 48.87| 62.04| 60.44| 34.13| 25.6|
> |unstructured + our prompt (50% sparsity)|**57.25**|**79.69**| **73.12**| **50.58**| **64.33**| **66.2**|**36.86**|**30.0**|
>
>
> |Model | Avg | BoolQ| RACE-High|
> | :--:   | :--:    |:--:        |:--:        |
> |2:4 sparsity| 51.59| 67.00| 36.17|
> |2:4 sparsity + our prompt| **53.55**| **69.11**| **37.99**|
> |unstructured (50% sparsity)| 52.16| 67.19| 37.13|
> |unstructured + our prompt (50 sparsity)| **54.09**| **70.95**| **37.22**|
>
> [1] SparseGPT: Massive Language Models Can Be Accurately Pruned in One-Shot

---

> ### Author Response · Authors · 2023-11-20
> **Author Response to Official Review by Reviewer dFpf (3/3)**
>
> > Incorporating the system pruning prompt during the inference phase can increase inference costs.
>
> **Response**: Thank you for your insightful question.  Naively incorporating the system pruning prompt during the inference phase can indeed increase the inference latency. However, this pruning prompt is a fixed prompt, which only accounts for 256 tokens for LLaMA models,  and applied to all inference requests. Thus, many strategies can be applied to reduce the latency overhead caused by this fixed prompt.
>
> For instance, we can simply leverage the key-value (KV) Cache approach, which eliminates the need to compute full attention for every token. To demonstrate this,  we take vllm framework[1] as an example which already supports the KV cache implementation. Specifically, we measure the inference latency of our pruned LLMs based on vllm framework on an A100. We randomly select 10 questions from the GPT4Alpaca dataset, and measure the inference latency of the generation process both with and without the use of the pruning prompt. The results are shown in the Table below. All latency numbers represent end-to-end latency. Our pruned LLMs of different sizes all accelerate the token generation process of the original LLaMA-7B, both with and without the pruning prompt. Specifically, Compresso-5.4B and 4.5B accelerate the generation process by 1.17x and 1.19x, respectively. When we add our pruning prompt, the latency slightly increases, but compared to LLaMA-7B, there is still an acceleration of 1.10x and 1.13x.
>
> | Model | Latency  | Acceleration|
> | :--:   | :--:    |:--:    |
> |LLaMA-7B| 195.58 ms| 1x|
> |Compresso-5.4B| 170.76 ms| **1.17x**|
> |Compresso-5.4B (Inference with Pruning Prompt)| 180.81 ms| **1.10x** |
> |Compresso-4.5B| 167.93 ms | **1.19x**  |
> |Compresso-4.5B (Inference with Pruning Prompt)| 177.16 ms| **1.13x** |
>
> Moreover, a recent work called "PromptCache" [2] has been introduced, which can be utilized to further mitigate the additional inference costs caused by the pruning prompt. PromptCache can pre-compute and reuse the attention states across multiple requests, which can significantly reduce the inference latency caused by our fixed system prompt. The latency reduction in the generation phase can be reduced by up to 10x, which is very promising. However, since it currently has not been open-sourced, we leave the latency measurement as future work.
>
> [1] https://github.com/vllm-project/vllm
>
> [2] Prompt Cache: Modular Attention Reuse for Low-Latency Inference https://arxiv.org/pdf/2311.04934.pdf

---

### Official Review · Reviewer_FZK6 · 2023-11-01

**Soundness:** 2 fair
**Presentation:** 3 good
**Contribution:** 2 fair
**Rating:** 5
**Confidence:** 3

**Summary:**

This work proposes a structured pruning method for large language models (LLMs) named Compresso. The work leverages the LoRA to achieve efficient training-based pruning. The work adopts $L_0$ reparameterization to enable differentiable masks to be jointly optimized with learnable parameters. Moreover, the work tries to prune with the proposed prompt to enhance the pruning results. The experimental result shows that Compresso can prune LLaMA-7B to a 5.4B size while maintaining the performance on zero-shot commonsense reasoning and reading comprehension, as well as few-shot MMLU and BBH benchmarks.

**Strengths:**

- The writing is clear and easy to understand.
- This research focuses on structured pruning for Language Model Models (LLMs), which can reduce the cost of making predictions without requiring specialized hardware support.
- The research employs many techniques to make the training-based pruning effective and reduce its costs.

**Weaknesses:**

- The work lacks novelty as many parts are taken from existing works such as LoRA and differentiable masks.
- The proposed collaborative prompt is interesting, but the empirical study fails to support it convincingly. The ablation study in Table 6 is inconsistent, and it is recommended to perform the Compresso without prompting on both training and inference. Additionally, the lack of analysis or discussion on collaborative prompts prevents readers from fully understanding it.
- The main experiment only includes one baseline, which makes the results less convincing.
- There needs to be more explanation or discussion of the performance improvement without post fine-tuning in Table 7.
- Equation 3 contains many undefined symbols and requires further clarification.

**Questions:**

Please refer to the weaknesses.

---

> ### Author Response · Authors · 2023-11-20
> **Author Response to Official Review by Reviewer FZK6 (1/3)**
>
> We appreciate the time and effort you put in reviewing our work, as well as your detailed comments and valuable questions. We understand the concerns you raised and we're pleased to address these concerns. We also appreciate the opportunity to provide some clarification.
>
> > Add more baselines
>
> **Response**: We are grateful for your suggestion to include more baselines for comparison. However, due to constraints such as the absence of open-source code or the requirement of substantial GPU resources, we have selected two one-shot baselines, SparseGPT[1] and Wanda [2], for comparison. They are originally designed for unstructured pruning, and can be extended for N:M sparsity. We further extend them to structured pruning, allowing the removal of entire attention heads and FFN intermediate size. All other settings follow the original papers.
>
> The results of our comparison are presented in the table below. More detailed numbers can be found in the updated PDF. Interestingly, we found that while SparseGPT and Wanda outperform LLM-Pruner on three of the benchmarks, they fall short on commonsense reasoning. Wanda performs better than SparseGPT at lower sparsity levels, but experiences a significant drop in accuracy at higher sparsity ratios (4.5B). One possible explanation is that Wanda relies on the activation and weights to conduct direct weight pruning, unlike SparseGPT which is dependent on second-order Hessian inverses and includes weight updates on pruned LLMs to recover information loss.
>
> In contrast, our method significantly outperforms these baselines across all benchmarks, suggesting that our method is the most effective at preserving the generalization capability of the original LLaMA-7B.
>
> | Model | Method    | Commonsense Reasoning (0-shot)      | Reading Comprehension (0-shot) | MMLU (5-shot) | BBH (3-shot)|
> | :--:   | :--:    |:--:        |:--:        |:--:        |:--:        |
> |5.4B| *SparseGPT* | 58.12|54.32 |26.80 | 27.83|
> | | *Wanda*| 58.06|55.54 | 24.50|28.96 |
> | | LLM-Pruner | 59.14|48.92 | 24.86| 28.97|
> | | **Compresso** | **60.09**|**60.35**| **31.90**|**31.47**|
> |5.0B| *SparseGPT* |55.18 |51.35 |25.21 |29.40 |
> | | *Wanda*|54.45 | 51.43| 23.32|24.86 |
> | | LLM-Pruner | 56.37| 48.93| 23.22| 26.46|
> | | **Compresso** |**57.05** |**56.58** |**27.68** |**31.27**|
> |4.5B| *SparseGPT* |52.27 |50.39 | 22.30| 26.61|
> | | *Wanda*|  34.33| 39.47| 15.65|16.08|
> | | LLM-Pruner | 53.73| 47.70| 23.85|24.67 |
> | | **Compresso** |**55.94** |**52.52**|**25.92**|**28.25** |
>
> [1] SparseGPT: Massive Language Models Can Be Accurately Pruned in One-Shot
>
> [2] A Simple and Effective Pruning Approach for Large Language Models

---

> ### Author Response · Authors · 2023-11-20
> **Author Response to Official Review by Reviewer FZK6 (2/3)**
>
> > The ablation study in Table 6 is inconsistent, and it is recommended to perform the Compresso without prompting on both training and inference.  More analysis or discussions for pruning prompt.
>
> **Response (1/2)**: Thank you for your constructive suggestion. We have corrected the typos in Table 6 for consistency. Additionally, following your suggestion, we conduct an experiment to further evaluate the effectiveness of Compresso, where the pruning prompt is removed from the entire process, including training, fine-tuning and inference. The Table below presents the performance of 5.4B pruned LLMs under different settings.
>
> |Task| Compresso with Prompting | No Prompt in Inference| No Prompt in Training| *No Prompt in Training and Inference*|
> | :--:   | :--:    |:--:        |:--:        | :--:    |
> |Commonsense Reasoning (0-shot) |60.09 | 56.07|57.36  | 57.09 |
> |Reading Comprehension (0-shot) |60.35 |56.52 |56.62 | 55.84|
> |MMLU (5-shot)| 31.90| 31.49|29.30 |29.10  |
> |BBH (3-shot) | 31.47|30.57 | 28.08| 28.64|
>
> We observe three key findings as follows:
>
> 1) The results align with the original paper, indicating that removing the pruning prompt at any stage reduces the performance of pruned LLMs.
> 2) We found that the impact of pruning prompt at different stages varies across benchmarks. For instance,  removing the pruning prompt at any stage affects zero-shot commonsense reasoning and reading comprehension. However, removing the pruning prompt during training has a greater impact on MMLU and BBH than during inference.
> 3) Even if the pruning prompt is removed from training, reintroducing it during inference can still improve the performance of tasks such as commonsense reasoning, reading comprehension, and MMLU, while it may decrease performance on BBH.
>
> In summary, the additional ablation study further validate the effectiveness of our collaborative pruning paradigm. The experiment results highlight the significant role of our collaborative pruning prompt during the training phase. Specifically, the integration of the pruning prompt during the training-based pruning process is pivotal, fostering a collaborative learning approach where both the LLM and the pruning algorithm jointly optimize for better pruning decisions.
>
> **Response (2/2)**: Moreover, we would like to share more empirical experiments to validate the effectiveness of our pruning prompt. Following the suggestion by Reviewer dFpf, we broaden our experimental scope to include other sparse patterns such as unstructured and N:M sparse patterns. **We demonstrate that our pruning prompt can also improve the pruning performance under unstructured and N:M sparsity patterns**.
>
> We utilize SparseGPT[1] for the experiment, which is a representative method for unstructured pruning and n:m sparsity. Specifically, we prune LLaMA-7B under a 50% sparsity ratio for unstructured pruning and under n:m (2:4) sparsity using SparseGPT. We then evaluate the pruned model under conditions with and without our pruning prompt for zero-shot commonsense reasoning and reading comprehension. As shown in the tables below, our pruning prompt significantly improves the performance on all benchmarks and tasks.
>
> |Model | Avg | StoryCloze| PIQA| HellaSwag| WinoGrande| ARC-e|ARC-c|OBQA|
> | :--:   | :--:    |:--:        |:--:        |:--:        |:--:        |:--:        |:--:        |:--:        |
> |2:4 sparsity | 51.46| 75.31| 70.18| 45.52| 58.33| 55.85| 31.4| 23.6|
> |2:4 sparsity + our prompt| **53.97**| **76.16**| **71.0**| **47.56**| **60.85**| **59.3**| **33.7**| **29.2**|
> |unstructured (50% sparsity)| 54.59| 79.10| 71.93| 48.87| 62.04| 60.44| 34.13| 25.6|
> |unstructured + our prompt (50% sparsity)|**57.25**|**79.69**| **73.12**| **50.58**| **64.33**| **66.2**|**36.86**|**30.0**|
>
> |Model | Avg | BoolQ| RACE-High|
> | :--:   | :--:    |:--:        |:--:        |
> |2:4 sparsity| 51.59| 67.00| 36.17|
> |2:4 sparsity + our prompt| **53.55**| **69.11**| **37.99**|
> |unstructured (50% sparsity)| 52.16| 67.19| 37.13|
> |unstructured + our prompt (50 sparsity)| **54.09**| **70.95**| **37.22**|
>
> [1] SparseGPT: Massive Language Models Can Be Accurately Pruned in One-Shot

---

> ### Author Response · Authors · 2023-11-20
> **Author Response to Official Review by Reviewer FZK6 (3/3)**
>
> > The work lacks novelty as many parts are taken from existing works such as LoRA and differentiable masks.
>
> **Response**: We thank you for your comment and appreciate the opportunity to clarify our contributions:
>  1. Our primary contributions are the introduction of the collaborative pruning LLM paradigm and the associated pruning prompt. Unlike conventional approaches treating LLMs passively, our paradigm makes the LLM an active participant in the pruning process, resulting in a substantial performance improvement. The behind insight is that recent state-of-the-art LLMs already have the capability to understand and adapt to complex operational instructions (i.e., encourage the LLM itself better adapt to the pruning process). We are pleased to note that Reviewers dFpf and VBbn have acknowledged and appreciated the novelty of this paradigm.
>  2. Besides, our structured pruning algorithm is not a simple combination of existing techniques. In contrast, every design choice is purposeful and aimed at addressing specific challenges. For instance, regarding LoRA, typically applied in fine-tuning and instruction tuning, we innovatively integrate it into the pruning process. This integration allows us to significantly reduce the resources required for training-based pruning while maintaining promising pruning performance.  The use of differentiable masks in our pruning algorithm aligns perfectly with our proposed collaborative pruning prompt. This is a departure from existing pruning algorithms that rely on weight importance metrics to determine which weights to prune, a process that presents challenges when integrating with our collaborative pruning prompt. This optimally leverages the capabilities of both the pruning algorithm and the LLM itself to learn the optimal pruning decisions.
>
> Thank you for your consideration, and we look forward to your feedback.
>
> > There needs to be more explanation or discussion of the performance improvement without post fine-tuning in Table 7.
>
> **Response**: Thank you for your valuable suggestion. Upon delving into the detailed numbers, we found that the slight performance decrease for the 4.5B model after fine-tuning is due to the less significant improvement on our proposed collaborative pruning prompt. As shown in the table below, we list out the BBH scores both with and without the collaborative pruning prompt. *The numbers in ( ) denote the BBH scores without the collaborative pruning prompt*. We observed that the average BBH, BBH on NLP, and BBH on algorithmic all improve after fine-tuning without the collaborative prompt, which aligns with common practice.  However, after fine-tuning, the score of BBH on algorithmic with the prompt only improves by 0.97, while it originally improves by 3.81 before the fine-tuning.
>
> | 4.5B on BBH | BBH Avg     | NLP      | Algorithmic |
> | :--:   | :--:    |:--:        |:--:        |
> | No Fine-tuning | 29.56 (27.65)| 32.29 (32.90)| 27.37 (23.56)|
> | After Fine-tuning| 28.25 (27.71) |32.62 (32.50) | 24.75 (23.78)|

---

> ### Comment · Reviewer_FZK6 · 2023-11-23
>
> After reviewing the feedback from the authors, I have decided to change my initial rating from 3 to 5 as the authors have addressed some of my concerns, but the novelty of the work is limited and needs more insight.

---

> ### Author Response · Authors · 2023-11-23
> **Official Comment by Authors**
>
> We sincerely appreciate your review and valuable feedback. We are pleased to see that our efforts to address your concerns have resulted in an improved evaluation of our paper. Your willingness to reconsider and increase the rating from 3 to 5 is sincerely appreciated.
>
> We would like to take this opportunity to reiterate and clarify our key contributions:
>
> 1. **Structured pruning of LLMs in training-based methods**.
> Our work sets itself apart from the majority of existing LLM pruning techniques, which predominantly focus on one-shot post-training pruning. These methods excel in unstructured and n:m sparsity pruning, but they fall short when it comes to structured pruning due to the pruning of entire channels/heads, leading to significant information loss. Compresso, on the other hand, pioneers the introduction of structured pruning within a training-based framework. We tackle the raised challenges of high training costs and data collection. Furthermore, since we jointly optimize the original model parameters and the pruning masks, we can better recover the errors caused by structured pruning, and the resulting structured pruned LLMs retain the original LLM capacity.
>
>
> 2. **Introduction of collaborative pruning paradigm and the pruning prompt**.
> We introduce a novel paradigm that harnesses the LLM in a collaborative pruning process. While we acknowledge the limited theoretical analysis in the current version, **the extensive experiments on different sparse patterns (i.e., unstructured pruning, n:m sparsity and structured pruning) and tasks demonstrate its effectiveness and unique role in LLM pruning**.  We are confident that this collaborative pruning paradigm and pruning prompt represents a promising direction for LLM pruning research. We are grateful for the recognition of its novelty by other reviewers (*dFpf and VBbn*), and we are enthusiastic about the potential it unveils for the LLM pruning community.
>
>
> 3. **Promising pruning results achieved with minimal training resources**.
> Our pruning experiments have yielded promising results. For example, we have demonstrated the ability to prune LLaMA-7B to 5.4B, LLaMA-13B to 11B, while preserving their original generalization capability. Moreover, this is achieved with minimal training resources, i.e., it requires only 4x 32GB V100 GPUs and an instruction-tuning dataset with 52k examples. Finally, the pruned LLMs achieve direct inference latency speedups. This stands in stark contrast to current unstructured and n:m semi-structured pruning methods, which necessitate specialized inference frameworks and hardware for comparable speedups.
>
>
> We once again express our gratitude for your time and consideration, and we eagerly anticipate any additional insights or suggestions you may have.

---

### Official Review · Reviewer_uGct · 2023-11-04

**Soundness:** 2 fair
**Presentation:** 3 good
**Contribution:** 3 good
**Rating:** 5
**Confidence:** 4

**Summary:**

This paper proposes a one-shot structured pruning method, called Compresso, for Large Language Models (LLMs). Compresso incorporates Low-Rank Adaptation (LoRA) into the L0 regularization during the instruction tuning process, and it also fosters collaboration between the LLM and the pruning algorithm. The evaluation shows that Compresso outperforms LLM-Pruner on several datasets.

**Strengths:**

1. The writing is clear.
2. The authors evaluated 7 benchmarks to show that Compresso outperforms LLM-Pruner.

**Weaknesses:**

1. In Equation 3, what $d_h$ denotes is not explained;  Similarly, In Equation 5, what $\lambda1$ and $\lambda2$ denote is not explained.
2. Some references or empirical evidence is needed to support it is the common practice of setting l to -0.1, r to 1.1, and β to 2.
3. In Section 4, it would be better if the authors could provide some insights, rather than only listing the numbers.
4. Demonstrating the theoretical underpinnings of Compresso would greatly enhance the paper.
5. This paper claims that LLM-Pruner is the only LLM structured pruning work. However, this claim appears to be open to question. The related works [1-7] that structurally prune LLMs are not mentioned or compared.
6. There are some typos in the paper, e.g. “We introduce LoRA intro pruning in a novel manner. Formally”.

[1] Graph-to-Text Generation with Dynamic Structure Pruning

[2] Sheared LLaMA: Accelerating Language Model Pre-training via Structured Pruning

[3] ZipLM: Inference-Aware Structured Pruning of Language Models

[4] Dynamic Sparse No Training: Training-Free Fine-tuning for Sparse LLMs

[5] A Simple and Effective Pruning Approach for Large Language Models

[6] Gradient-Free Structured Pruning with Unlabeled Data

[7] LoRAShear: Efficient Large Language Model Structured Pruning and Knowledge Recovery

**Questions:**

1. Is it possible to compare Compresso with the related works (mentioned above)?
2. In Section 3.3, the authors claim that this paper introduces LoRA into pruning in a novel manner”. Can you explain the main difference between LoRA used in fine-tuning LLMs and the way used in pruning in this paper?
3. Although Compresso is a one-shot pruning method, can the performance of LLMs be improved if we do Compresso iteratively?
4. How to adjust the hyperparameters λ1 and λ2 using the AdamW optimizer? Can authors provide more details or an ablation study about that?

---

> ### Author Response · Authors · 2023-11-20
> **Author Response to Official Review by Reviewer uGct (1/4)**
>
> We greatly appreciate your time, detailed comments, and valuable suggestions. We are pleased to address your concerns and make some clarifications.
>
> > Compare Compresso with related works
>
> **Response (1/2): Clarifications on the listed related works**:
> We greatly appreciate your insightful comments and the opportunity to clarify our work in relation to the cited works.
>
> Firstly, we would like to address the timing of the publication of some of LLM structured pruning works, such as [2][7]. These papers were published on Arxiv after ICLR deadline, specifically after October 10. At the time of our paper's submission, we were not aware of any other structured pruning works on state-of-the-art LLMs like LLaMA, apart from LLM-pruner. Consequently, we stated in our paper that LLM-pruner was the only LLM structured pruning work. We have revised this statement in the updated PDF and added the appropriate citations.
>
> Although [2][7] are concurrent works, we would like to highlight the differences between our work and [2][7] as follows:
> Sheared LLaMA [2] utilizes RedPajama, a replicated pretraining dataset for LLaMA with 50B tokens, for both pruning and fine-tuning. However, it requires 16 A100s with 80GB of memory[2], which may not be feasible for public users. In contrast, our method uses an instruction tuning dataset with 52k instances and integrates LoRA in pruning. This allows for experiments on 4x 32GB V100s, a more efficient and practical approach acknowledged by Reviewer VBbn. Apart from its pruning dataset, the pruning algorithm of Sheared LLaMA does not present much novelty, as it only introduces minor modifications to the original CoFiPruning[8]. In contrast, our method proposes a new learning-based structured pruning paradigm, which leverages the LLM itself to conduct collaborative pruning, this differentiates our work from traditional pruning works.
>
> LoRAShear[7] is a one-shot structured pruning work. It initially creates dependency graphs over LoRA modules and analyzes the knowledge distribution. It then iteratively prunes based on this distribution. However, due to the unavailability of its source code and the limited time for rebuttal, we have not included it in our comparison.
>
> Secondly, some of the related works you listed, like [4][5], are semi-structured pruning (n:m sparsity) that requires specified inference runtime and hardware for acceleration, while our focus is entirely on structured pruning that directly removes individual attention heads and reduces FFN intermediate size. [1] is specifically designed for dynamically pruning the structure of graph data, not on the pre-trained language model, while [6] is for BERT, not LLM. [3] only evaluated on BERT and GPT/GPT2. And there is no available open-source code for [3] and [6].
>
> We hope this clarifies our position and the unique contributions of our work. We look forward to further discussions and feedback.
>
> [1] Graph-to-Text Generation with Dynamic Structure Pruning
>
> [2] Sheared LLaMA: Accelerating Language Model Pre-training via Structured Pruning
>
> [3] ZipLM: Inference-Aware Structured Pruning of Language Models
>
> [4] Dynamic Sparse No Training: Training-Free Fine-tuning for Sparse LLMs
>
> [5] A Simple and Effective Pruning Approach for Large Language Models
>
> [6] Gradient-Free Structured Pruning with Unlabeled Data
>
> [7] LoRAShear: Efficient Large Language Model Structured Pruning and Knowledge Recovery.
>
> [8] CoFiPruning: Structured Pruning Learns Compact and Accurate Models

---

> ### Author Response · Authors · 2023-11-20
> **Author Response to Official Review by Reviewer uGct (2/4)**
>
> > Compare Compresso with related works
>
> **Response (2/2): Comparision results with two new baselines**:  We are grateful for your suggestion to include more baselines for comparison. However, due to constraints such as the absence of open-source code or the requirement of substantial GPU resources, we have selected two one-shot baselines, SparseGPT and Wanda [5], for comparison. They are originally designed for unstructured pruning and can be extended to N:M sparsity. We further extend them to structured pruning, allowing the removal of entire attention heads and FFN intermediate size. All other settings follow the original papers.
>
> The results of our comparison are presented in the Table below. More detailed numbers can be found in the updated PDF. Interestingly, we found that while SparseGPT and Wanda outperform LLM-Pruner on three of the benchmarks, they fall short on commonsense reasoning. Wanda performs better than SparseGPT at lower sparsity levels, but experiences a significant drop in accuracy at higher sparsity ratios (4.5B). One possible explanation is that Wanda relies on the activation and weights to conduct direct weight pruning, unlike SparseGPT which is dependent on second-order Hessian inverses and includes weight updates on pruned LLMs to recover information loss.
>
> In contrast, our method significantly outperforms these baselines across all benchmarks, suggesting that our method is the most effective at preserving the generalization capability of the original LLaMA-7B.
>
> | Model | Method    | Commonsense Reasoning (0-shot)      | Reading Comprehension (0-shot) | MMLU (5-shot) | BBH (3-shot)|
> | :--:   | :--:    |:--:        |:--:        |:--:        |:--:        |
> |5.4B| *SparseGPT* | 58.12|54.32 |26.80 | 27.83|
> | | *Wanda*| 58.06|55.54 | 24.50|28.96 |
> | | LLM-Pruner | 59.14|48.92 | 24.86| 28.97|
> | | **Compresso** | **60.09**|**60.35**| **31.90**|**31.47**|
> |5.0B| *SparseGPT* |55.18 |51.35 |25.21 |29.40 |
> | | *Wanda*|54.45 | 51.43| 23.32|24.86 |
> | | LLM-Pruner | 56.37| 48.93| 23.22| 26.46|
> | | **Compresso** |**57.05** |**56.58** |**27.68** |**31.27**|
> |4.5B| *SparseGPT* |52.27 |50.39 | 22.30| 26.61|
> | | *Wanda*|  34.33| 39.47| 15.65|16.08|
> | | LLM-Pruner | 53.73| 47.70| 23.85|24.67 |
> | | **Compresso** |**55.94** |**52.52**|**25.92**|**28.25** |

---

> ### Author Response · Authors · 2023-11-20
> **Author Response to Official Review by Reviewer uGct (3/4)**
>
> > How to adjust the hyperparameters λ1 and λ2 using the AdamW optimizer? Can authors provide more details or an ablation study about that?
>
> **Response**: Thank you for the question! We have two AdamW optimizers in our pruning process. The first optimizer is used for optimizing the inserted LoRA modules for updating model parameters, while the second is used to learn differentiable masks. Specifically, λ1 and λ2 are optimized in conjunction with the second AdamW optimizer and the masks. These parameters, as gradient-enabled NN.parameters, are passed to the second AdamW optimizer. Given our initial reglr=0.05, they work together with the first AdamW optimizer to learn the best pruning decision. The value of 'reglr' can impact λ1, λ2, and the overall pruning performance. Specifically, a larger 'reglr' tends to optimize the Lagrangian loss in Equation 5, ensuring that the model sparsity of the pruned LLM strictly reaches the target sparsity ratio during the entire training process. Conversely, a smaller 'reglr' makes the pruning process more inclined to optimize the model prediction loss, but it may not guarantee a well-aligned sparsity ratio during the entire pruning process.
>
> To validate this, we conduct an ablation study. As shown in the table below, in our original paper's experiment, we empirically set 'reglr' to 0.05. Now we chose a larger 'reglr' of 0.1 and a slightly smaller 'reglr' of 0.02. These three different 'reglr' values all can ensure that the model sparsity reaches the target sparsity ratio after pruning. However, both a large and small 'reglr' can impact the pruning performance. In our experiments, we empirically set 'reglr' to 0.05, and it performs well on both LLaMA-7B and 13B.
>
>
> |λ1, λ2| Commonsense Reasoning (0-shot) | Reading Comprehension (0-shot)|
> | :--:   | :--:    |:--:        |
> |0.1 | 56.10|56.13 |
> |0.05 (used in our experiments)| 60.09|60.35 |
> |0.02 | 58.54 | 56.36 |

---

> ### Author Response · Authors · 2023-11-20
> **Author Response to Official Review by Reviewer uGct (4/4)**
>
> > Although Compresso is a one-shot pruning method, can the performance of LLMs be improved if we do Compresso iteratively?
>
> **Response**: Thank you for your thoughtful consideration and question! I appreciate your question, but there seems to be slight misunderstanding about Compresso. Contrary to the assumption that it is a one-shot pruning method, Compresso employs a collaborative pruning paradigm and differentiable masks. This allows it to learn optimal pruning decisions, making it more than a traditional one-shot method. I hope this clarification provides a clearer understanding of Compresso and its capabilities. Please let us know if you have any further questions.
>
> > Demonstrating the theoretical underpinnings of Compresso would greatly enhance the paper
>
> **Response**: Thanks for your suggestion! We acknowledge that theoretical analysis can provide deeper insight into our proposed pruning process. However, the behavior of LLMs remains unclear and is still an open question in the literature, making it challenging to conduct a theoretical analysis of our proposed pruning paradigm, especially regarding why our collaborative pruning prompt works. Instead, we conduct extensive experiments to demonstrate the effectiveness and generalizability of our approach. Our experiments span diverse datasets and multiple models under different sparsity ratios (new results on LLaMA13b can be found in the Appendix A.1), and show very promising results. We leave the theoretical analysis as the future works.
>
>
> > Can you explain the main difference between LoRA used in fine-tuning LLMs and the way used in pruning?
>
> **Response**: We appreciate your attention and have updated the PDF to address any confusion. Our initial goal was to emphansize that our method of optimizing differentiable masks naturally integrates with LoRA, as shown in equation (2). The implementation of LoRA used in fine-tuning and the way used in pruning in this paper are very similar, with the only differences being that both masks and LoRA modules are optimized together during pruning.
>
> However, we would like to emphasize that in the context of structured pruning based on training, traditional pruning methods and the recent Sheared LLaMA all require full gradient updates on the model parameters, which necessitate substantial training and GPU resources. By using LoRA to allow the update of a few model parameters during pruning, we can significantly reduce the training costs. For instance, we are able to run LLama13b on 4 V100 GPUs. However, this may also impact the pruning performance as only a few model parameters are allowed to update. Our Compresso demonstrates that it achieves superior performance despite these constraints.

---

### Author Response · Authors · 2023-11-20
**General Response to All Reviewers and ACs**

We greatly thank all reviewers and ACs for their time and efforts in reviewing our paper and giving insightful comments. We are pleased that the reviewers have recognized our contributions list below:

1) **Novelty**. We introduce a new collaborative pruning paradigm for LLM. This involves a training-based pruning algorithm that works together with the LLM using a pruning prompt to **learn the optimal pruning decisions**. This distinguish this work from existing pruning methods (dFpf and VBbn).
2) **Pruning efficiency**. Our approach's outstanding efficiency allows for training and pruning LLaMA-7B and 13B using only 4 Nvidia V100 GPUs (FZK6 and VBbn). This is a significant contribution, especially when compared to other training-based pruning methods like Sheared LLaMA [1], which requires 16 A100 GPUs for pruning and fine-tuning, making it less accessible to many users.
3) **Practical applicability**. Unlike most LLM pruning works focus on unstructured or N:M weight prunig, we conduct structured pruning, which can reduce LLM inference cost without requiring specialized hardware support    (FZK6 and VBbn)
4) **Experiments across multiple domains**. We set a new standard in the pruning of LLMs by outperforming one-shot pruning methods on 4 domain benchmarks (uGct and VBbn). While existing pruning baselines often evaluate and report good performance on perplexity and commonsense reasoning benchmarks, *they have not demonstrated the generalization capability of their pruned LLMs. As we found, they often fall short on other domain benchmarks such as reading comprehension, few-shot MMLU, and BBH*. In contrast, our method consistently outperforms these baselines across all benchmarks, demonstrating its superior ability to preserve the generalization capability of the original LLMs.


We also appreciate the constructive suggestions and concerns raised by the reviewers, which have spurred a productive and informative discussion and additional experiments. We have made appropriate revisions, with all changes marked in purple for easy reference. We believe these revisions further strengthen our paper. We summarize our major changes as follows:

1) Add discussion on concurrent related works (Section 2)
2) Add two additional comparison baselines (Section 4.2)
3) Add more analysis in Section 4 (Section 4.2)
4) Add more experiments to further demonstrate the effectiveness of the collaborative pruning paradigm and the proposed prompt (Section 4.3)
5) Refine the writing, adjusting the tone of some sentences to be more conservative, and provide more detailed explanations for Equation notations.
6) Add pruning results for LLaMA-13B (In Appendix A.1)
7) Detailed model structures of the pruned LLM (In Appendix A.1)
8) Add additional experiments to demonstrate that our proposed collaborative pruning prompt can be generalized on other sparse patterns (i.e., unstructured and n:m weight pruning scenarios) (In Appendix A.1)
9) Add latency measurement to demonstrate the real inference acceleration (In Appendix A.1)
10) Add a section discussing the limitations of our current work and potential directions for future research (In Appendix A.2)


Finally, we have **addressed all the raised questions and concerns**, and we present detailed responses to individual reviewers below.

[1] Sheared LLaMA: Accelerating Language Model Pre-training via Structured Pruning

---

### Author Response · Authors · 2023-11-22
**A kind reminder regarding our response**

Dear Reviewers,

We have put in many efforts to address the raised questions and concerns. As the ICLR rebuttal period is approaching its end, we kindly remind you to review our submitted response. Your feedback is essential for finalizing our work.

Thank you for your attention.

Best regards,

The Authors

---

### Meta-Review · Area_Chair_RGsf · 2023-12-10

**Metareview:**

This paper proposes a structure pruning method for llms, which utilizes a prompt to improve the performance of the pruned model. The proposed method is shown to outperform existing methods such as LLM-Pruner, and the idea of collaborative prompting is interesting. However, reviewers and AC are leaning towards rejection due to lack of comparisons to existing baselines, potential unfair experiments for using stronger instruction tuning datasets, and insufficient justification and analysis of the proposed collaborative prompting approach.

**Justification For Why Not Higher Score:**

lack of comparisons to existing baselines, potential unfair experiments for using stronger instruction tuning datasets, and insufficient justification and analysis of the proposed collaborative prompting approach

**Justification For Why Not Lower Score:**

N/A

---

### Decision · Program_Chairs · 2024-01-16

Reject